# Evaluation of tropospheric water vapour and temperature profiles retrieved from Metop-A by the Infrared and Microwave Sounding scheme

**Tim Trent**[1,2], **Richard Siddans**[3,4], **Brian Kerridge**[3,4], **Marc Schröder**[5], **Noëlle A. Scott**[6], and **John Remedios**[1,2]

[1]Earth Observation Science, School of Physics and Astronomy, University of Leicester, Leicester, UK
[2]National Centre for Earth Observation, Department of Physics and Astronomy, University of Leicester, Leicester, UK
[3]RAL Space, Remote Sensing Group, Harwell Oxford, Chilton, UK
[4]National Centre for Earth Observation, Harwell Oxford, Chilton, UK
[5]Satellite-Based Climate Monitoring, Deutscher Wetterdienst/Frankfurter Strasse 135, 63067 Offenbach, Germany
[6]Laboratoire de Météorologie Dynamique, Ecole Polytechnique–CNRS, 91128 Palaiseau, France

**Correspondence:** Tim Trent (t.trent@le.ac.uk)

**Abstract.** Since 2007, the Meteorological Operational satellite (Metop) series of platforms operated by the European Organisation for the Exploitation of Meteorological Satellites (EUMETSAT) have provided valuable observations of the Earth's surface and atmosphere for meteorological and climate applications. With 15 years of data already collected, the next generation of Metop satellites will see this measurement record extend to and beyond 2045. Although a primary role is in operational meteorology, tropospheric temperature and water vapour profiles will be key data products produced using infrared and microwave-sounding instruments onboard. Considering the Metop data record that will span 40 years, these profiles will form an essential climate data record (CDR) for studying long-term atmospheric changes. Therefore, the performance of these products must be characterised to support the robustness of any current or future analysis. In this study, we validate 9.5 years of profile data produced using the Infrared and Microwave Sounding (IMS) scheme with the European Space Agency (ESA) Water Vapour Climate Change Initiative (WV_cci) against radiosondes from two different archives. The Global Climate Observing System (GCOS) Reference Upper-Air Network (GRUAN) and Analysed RadioSoundings Archive (ARSA) data records were chosen for the validation exercise to provide the contrast between global observations (ARSA) with sparser characterised climate measurements (GRUAN). Results from this study show that IMS temperature and water vapour profile biases are within 0.5 K and 10% of the reference for 'global' scales. We further demonstrate the difference between diurnal sampling and cloud amount matchups on observed biases and discuss the implications sampling also plays on attributing these effects. Finally, we present the first look at the profile bias stability from the IMS product, where we observe global stabilities ranging from -0.32±0.18 to 0.1±0.27 K/decade, and -1.76±0.19 to 0.79±0.83 % ppmv/decade for temperate and water vapour profiles respectively. We further break down the profile stability into diurnal and latitudinal values and relate all observed results to required climate performance. Overall, we find the results from this study demonstrate the real potential for tropospheric water vapour and temperature profile CDRs from the Metop series of platforms.

## 1 Introduction

The water cycle, the largest movement of any substance between the surface and atmosphere, is a critical component of the Earth climate system (Chahine, 1992). Most water resides in the ocean and land reservoirs (ice, snow, surface/underground water and biota); however, the small fraction (< 1 % by mass) found in the atmosphere acts as a greenhouse gas warming the lower atmosphere. As a greenhouse gas, water vapour has a predominant capacity for positive feedback of

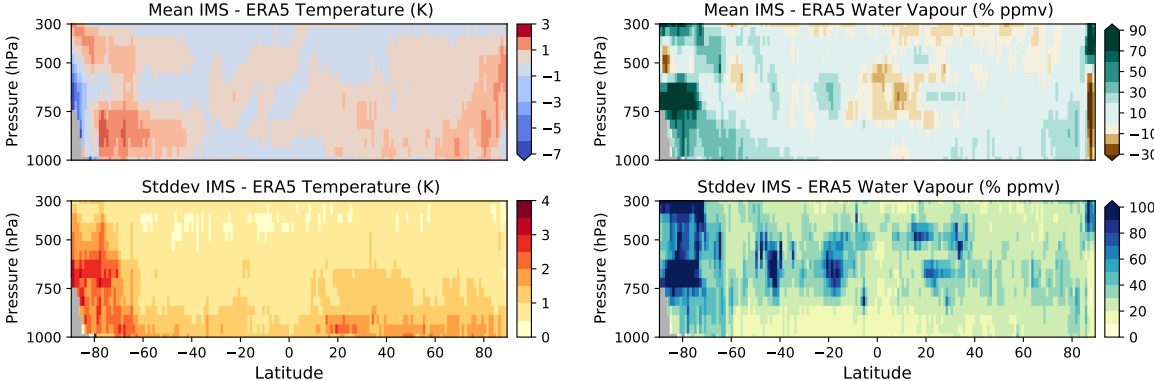

**Figure 1.** Example of the global mean differences between IMS temperature and water vapour profiles and ERA5 reanalysis for the $15^{th}$ June 2012. Also included are the standard deviations (stddev) for the differences. Reanalysis has been interpolated to the observation time and the centre of the IASI instantaneous field view. Before differences were calculated, the IMS averaging kernels were applied to the reanalysis profiles. Both IMS and ERA5 use IASI data. Therefore, differences are partly due to the differing backgrounds (a priori) and the different information extracted from the satellite radiances. For further discussion on averaging kernels refer to Section 3 (Methodology).

approximately 2 W m$^{-2}$ K$^{-1}$ (Dessler et al., 2008), acting as a powerful amplification mechanism for anthropogenic climate change compared to radiative forcing from other greenhouse gases (Chung et al., 2014). Water vapour also influences (directly and indirectly) the radiative balance of the Earth as well as surface and soil moisture fluxes. However, it is also sufficiently abundant and short-lived that it is considered to be under natural control (Sherwood et al., 2010). In the troposphere (the lowest 8-12 km), water vapour concentrations vary by four orders of magnitude between i) the surface and the tropopause and ii) wet tropical and dry polar latitudes. This global distribution, along with high temporal variability, results in tropospheric water vapour playing a significant role in global climate to micrometeorology scale processes (Bevis et al., 1992). Therefore, accurately capturing distributions and changes in atmospheric water vapour is critical for climate studies (Held and Soden, 2000; Trenberth et al., 2005).

The capability for observing tropospheric water vapour has been around since 1966, with the Medium Resolution Infrared Radiometer (MRIR) flown on the Nimbus-2 platform (NASA:a). This instrument consisted of five channels, one sensitive to upper tropospheric humidity (UTH) operating in the 6$\mu$m region (NASA:b). Subsequent advances in MRIR have seen the instrument evolve into the High-resolution InfraRed Sounder (HIRS), in which the fourth (and final) generation (HIRS/4) is a nineteen-channel instrument operating between 3.76 and 14.95 $\mu$m in the mid-infrared. The first combination of HIRS with companion microwave (MW) instruments sensitive to temperature and humidity in 1979 onboard the Television and Infrared Observation Satellite (TIROS)-N mission (NOAA 4th generation series satellite prototype). This combination of instruments became known as the TIROS Operational Vertical Sounder (TOVS) configu-

ration (Smith et al., 1979). The TOVS setup was operated until May 2007 onboard the NOAA-6 to NOAA-14 missions. In 1998 the NOAA-15 satellite was launched with the Advanced Television and Infrared Observation Satellite Operational Vertical Sounder (ATOVS), consisting of the Advanced Microwave Sounding Units (AMSU-A and AMSU-B) and HIRS/3 (Li et al., 2000) provided significant improvements over TOVS, especially for Numerical Weather Prediction (NWP) (English et al., 2000). This technological change also allowed for course profiles of tropospheric humidity and temperature to be inferred operationally (Courcoux and Schröder, 2015). Finally, the launch of the Atmospheric Infrared Sounder (AIRS) in 2002 (Chahine et al., 2006) and the Infrared Atmospheric Sounding Interferometer (IASI) in 2006 (Hilton et al., 2012) allowed water vapour and temperature profiles to be retrieved with an increased vertical resolution. This capability will be maintained out into the 2040s through the Joint Polar Satellite System (JPSS) and the Meteorological operational satellite Second Generation (Metop-SG) programmes. With nearly 15 years in space, the current IASI series of instruments represent a Climate Data Record (CDR) in their own right.

This study evaluates a 9.5-year record of temperature and humidity profiles from IASI and its companion MW instruments onboard Metop A, retrieved using the RAL IMS scheme, developed through both UK and European Organisation for the Exploitation of Meteorological Satellites (EUMETSAT) funding and produced as part of the European Space Agency (ESA) Water Vapour Climate Change Initiative (WV_cci). While modern NWP systems assimilate some spectral information from IASI and other satellites, the IMS product is designed to be independent of reanalysis. Therefore, in addition to climate model evaluation, tropospheric profile information from IMS can be used for comparative

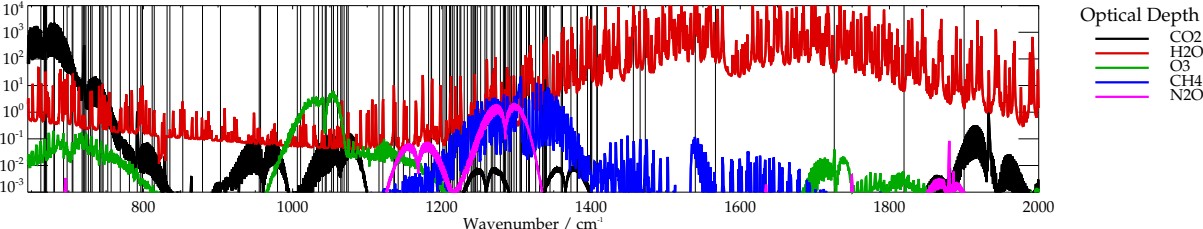

**Figure 2.** IASI channels used by the IMS scheme (indicated by vertical black lines) and nadir optical depths of major absorbers.

studies of reanalysis for both meteorological and climate applications. An example of this application is shown in Figure 1, where ERA5 has been collocated with IMS water vapour and temperature profiles. Here we see the daily differences between the data from satellite and reanalysis, with the most significant differences observed over polar regions. The assertion here is that the IMS will look to maximise information content from each set of measurements in a way that is too computationally expensive for reanalysis. However, for users to be confident about using IMS in such a manner, profiles need to be validated so that their performance is characterised.

Here, the validation of the IMS data archive is done using two radiosonde archives; i) the GCOS Reference Upper-Air Network (GRUAN, Immler et al. (2010)), and ii) Analyzed RadioSoundings Archive (ARSA, Scott (2015)). Both records cover the whole study period, with GRUAN supplying characterised soundings with higher vertical resolution at selected climate sites and ARSA providing global coverage with coarser vertical resolution. This approach allows for localised performance to be compared to broader global results, enabling a thorough test of the applicability of the IMS data for use as a CDR for temperature and humidity. This paper is structured as follows: section 2.1 provides a detailed description of the IMS algorithm used to generate the IASI temperature and water vapour profiles, with the radiosonde records discussed in section 2.2. Methods used for collocation and analysis by this study are provided in section 3, with results presented in section 4.

## 2   Data

This section describes the algorithm used to retrieve water vapour and temperature profiles from IASI with companion MW sounder data and details of the radiosonde data sets used for their assessment.

### 2.1   IMS

The RAL Infrared Microwave Sounding (IMS) core scheme employs the optimal estimation method (OEM) to jointly retrieve profiles of water vapour, temperature and stratospheric ozone, along with surface spectral emissivity and cloud pa-rameters from the Infrared Atmospheric Sounding Interferometer (IASI), Microwave Humidity Sounder (MHS) and the Advanced Microwave Sounding Unit (AMSU) on the MetOp satellites. The addition of spectral emissivity and cloud parameters to the state vector improved on the agreement with the European Centre for Medium-Range Weather Forecasts (ECMWF) analyses of lower tropospheric water vapour. These developments also reduced the sensitivity to cloud contamination, significantly improving global coverage. Employing very weak constraints based on zonal mean climatologies of water vapour, temperature and ozone, IMS is independent (in practice) of profile information. Therefore, this makes the IMS profile data record ideal for climate studies.

#### 2.1.1   Algorithm Description

The IMS algorithm is described in detail in Siddans (2019). It uses optimal estimation (Rodgers, 2000) to fit a set of measurements (in measurement vector, $\mathbf{y}$) with known error covariance, $\mathbf{S_y}$, by optimising a set of retrieved parameters (in state vector, x) using a "forward model" (FM), $F(\mathbf{x})$, capable of predicting the measurements from an estimate of the state. This inverse problem is solved using an a priori estimate of the state, a, and its assumed error covariance, $\mathbf{S_a}$. The solution state is found by minimising the following cost function (in this case, using the Levenberg-Marquardt method):

$$\chi^2 = (\mathbf{y} - F(\mathbf{x}))^T \mathbf{S_y}^{-1}(\mathbf{y} - F(\mathbf{x})) + (\mathbf{x} - \mathbf{a})^T \mathbf{S_a}^{-1}(\mathbf{x} - \mathbf{a})$$

$$(1)$$

The IMS scheme uses RTTOV 10 (Saunders et al., 2012) as the primary radiative transfer model (inside the FM). The measurement vector contains a sub-set of IASI, AMSU and MHS spectral channels: For IASI, IMS follows the same approach as the V6 operational OEM scheme (EUMETSAT, 2014, 2017) to pre-process the measurements and describe their errors. In particular, IMS uses IASI L1C spectra, which have been compressed and re-constructed using the operational principal components (Atkinson et al., 2010), which tends to filter noise. A further filter is applied to remove other instrumental artefacts (Hultberg and August, 2017). The 139 IASI channels (between 662.5 and 1900 cm$^{-1}$) se-

lected by EUMETSAT via information content analysis (see Figure 2) are used by the retrieval algorithm. The v6 scheme used a scan-dependent spectral bias correction for IASI, determined by comparing observed spectra to RTTOV simulations, based on atmospheric profiles from the version 6 piecewise-linear regression (PWLR) scheme. The correction was parameterised as a function of the view zenith angle using two spectra, $b_0$ and $b_1$, to represent the mean bias spectrum and its (assumed) linear dependence on the secant of the view zenith angle. The measurement error covariance matrix was calculated from the differences between bias-corrected IASI measurements and the RTTOV calculations. The IMS scheme uses the same spectral selection, measurement covariance and bias correction spectra. However, instead of assuming a fixed view zenith angle dependence for the bias correction, IMS jointly retrieves two parameters, $x_{b0}$ and $x_{b1}$, which scale the spectra $b_0$ and $b_1$. The scaled spectra are added to the RTTOV simulation, $R(\mathbf{x})$ in the FM:

$$F(\mathbf{x}) = R(\mathbf{x}) - x_{b0} \cdot b_0 - x_{b1} \cdot b_1 \qquad (2)$$

The bias correction is needed to account for systematic differences between RTTOV and the IASI observations, including errors in RTTOV. Allowing the retrieval to fit scale factors $x_{b0}$ and $x_{b1}$ instead of assuming a fixed scan-angle dependence improves the fit (gives lower cost) over a wide range of observing conditions. Examples of these corrections to systematically biased spectra are given in Siddans and Gerber (2015). The recent study by Calbet et al. (2018) supports this approach as the authors demonstrated that the inhomogeneities in water vapour within a satellite Instantaneous Field of View (IFOV) cause a significant modification in the results from radiative transfer modelling. Observational inhomogeneities across the IASI IFOV are predominantly due to clouds within the scene. These effects are accounted for at the L1 data stage by EUMETSAT through the collocation of AVHRR images within the IASI IFOV (EUMETSAT, 2019).

For Metop-A, IMS uses all AMSU-A and MHS channels except for channels 7 and 8 due to instrumental problems. An across-track-dependent bias correction is applied to the AMSU and MHS measurements (fixed as a function of view zenith), based on analysis measurements and simulations for a set of cloud-free scenes over sea equatorward of 60 deg (Siddans and Gerber, 2015). The complete measurement vector comprises the selected IASI and microwave sounder measurements in a single vector. Errors in IASI channels are assumed to have no correlation with errors in microwave sounder channels.

The state-vector, $\mathbf{x}$, contains parameters representing surface temperature and emissivity, the temperature, water vapour and ozone profiles, cloud fraction, cloud height, and the bias correction scale factors, $x_{b0}$ and $x_{b1}$. The state vector elements (along with their a priori values and variances) are described in more detail below (for further information, see (Siddans, 2019)). The a priori covariance is diagonal. Temperature, water vapour and ozone profiles are represented using basis functions, $\mathbf{Mx}$, which are the leading Eigenvectors of a covariance matrix representing the prior variability of the profiles on the 101 RTTOV pressure levels. For temperature, 28 vectors are fitted, with 18 for water vapour and 10 for ozone.

The covariance matrices were determined by computing the differences from the zonal mean of ECMWF analysis profiles for three days (17 April, 17 July, and 17 October 2013). The zonal mean and covariance matrix were computed in K for temperature and ln(vmr) for water vapour and ozone. The state vector comprises the coefficients of the leading Eigenvectors of the covariance matrix: Temperature profiles in (K) on the 101 RTTOV pressure levels are defined (in the FM before calling RTTOV) from the corresponding 28 elements of the state vector as follows:

$$\mathbf{T} = \mathbf{m}_T(\lambda) + \mathbf{M}_T \mathbf{x}_T, \qquad (3)$$

Where $\mathbf{m}_T$ is the zonal mean (interpolated to the latitude of observation); $\mathbf{M}_T$ is the matrix of Eigenvectors and $\mathbf{x}_T$ the temperature sub-set of the state vector. Water vapour and ozone profiles (in ppmv) are defined similarly with an exponent:

$$\mathbf{w} = e^{\left(\mathbf{m}_W(\lambda) + \mathbf{M}_W \mathbf{x}_W\right)} \qquad (4)$$

The a priori state vector elements for temperature, water vapour and ozone are all zero (the zonal mean of each profile is added in the FM). The Eigenvalues of the covariance matrix are used as the a priori variances. A similar approach in the spectral domain is adopted to represent surface emissivity. The state vector includes weights for the 20 leading Eigenvectors of an assumed global spectral emissivity covariance. The covariance is constructed using the RTTOV emissivity atlases to simulate emissivity in all the channels of IASI, AMSU and MHS from the same set of scenes used to define the profile covariances. However, this approach is insufficient because only a limited amount of spectral information is represented in the RTTOV atlases. To accurately simulate spectra in all used IASI channels, it is necessary to introduce further spectral patterns from the University of Wisconsin emissivity database (Borbas and Ruston, 2010). Prior values for the emissivity weights are set based on the RTTOV atlas for the specific location. Eigenvalues of the global covariance are used to define the a priori variances.

Cloud is modelled (via RTTOV) as a black body. The area fraction and top pressure are both retrieved for the cloud fraction. The state vector is the natural logarithm of cloud fraction with a prior and first guess value ln(0.01) and a prior error of 10. The log representation is adopted to prevent negative values of cloud fraction from arising. For cloud top

height, the state vector is defined in terms of the cloud pressure, **p**:

$$\mathbf{z}^* = 16(3 - log_{10}\mathbf{p}), \qquad (5)$$

where $\mathbf{z}^*$ corresponds approximately to altitude, the a prior and first guess values are assumed to be 5 km with a priori error also 5 km.

Although not retrieved, variations in $CO_2$, $CH_4$ and $N_2O$ are represented by a monthly latitude-dependent climatology derived from the Monitoring Atmospheric Composition and Climate (MACC) greenhouse gas (GHG) flux inversion reanalysis (Bergamaschi et al., 2013). Surface pressure is defined from ECMWF analysis (ERA-Interim, Dee et al. (2011)), adjusted to the mean altitude within the IASI footprint, assuming the logarithm of the surface pressure varies linearly with the difference between the IASI altitude and that of the ECMWF model.

A simple brightness temperature difference (BTD) test is applied for each scene to detect optically thick and high-altitude clouds using the IASI observation at 950 cm$^{-1}$ and a simulation with ECMWF analysis. The scene is not processed if the BT difference (observation –simulation) is outside the range of -5 to 15K. Residual cloud will remain in a significant fraction of scenes; the joint retrieved cloud fraction and height allow this to be accommodated to some extent and can be used to more stringently cloud clear the retrievals.

IMS provides several diagnostics from the OE which can be used to characterise the retrieved quantities (Rodgers, 2000): The error covariance ($\mathbf{S}_x$) for a given solution using an optimal estimation retrieval framework is given by:

$$\mathbf{S}_x = \left(\mathbf{S}_a^{-1} + \mathbf{K}^T\mathbf{S}_y^{-1}\mathbf{K}\right)^{-1}. \qquad (6)$$

With the transformation from the state vector to vertical profiles within IMS being expressed as a matrix operation (equations 3 & 4), the corresponding error covariances for layer averages are obtained (e.g. for temperature) by:

$$\mathbf{S}_T = \mathbf{M}_T\mathbf{S}_{x:T}\mathbf{M}_T^T. \qquad (7)$$

Where $\mathbf{S}_{x:T}$ is the sub-matrix of the error covariance for the temperature elements only. Water vapour and ozone profiles require an additional conversion from log units to obtain the covariance of the mixing ratio profile in ppmv:

$$\mathbf{S}_q = (\mathbf{q}\mathbf{M}_q)\mathbf{S}_{x:q}(\mathbf{q}\mathbf{M}_q)^T \qquad (8)$$

Where $\mathbf{q}$ is the retrieved water vapour profile in ppmv. The averaging kernel ($\mathbf{A}$) can account for the vertical sensitivity of the retrieved state vector and the influence of the prior. This is because the averaging kernel characterises the sensitivity of the retrieved state to the actual state (e.g. for water vapour):

$$\mathbf{A}_{f:q} = \mathbf{G}\mathbf{K}_{f:q}. \qquad (9)$$

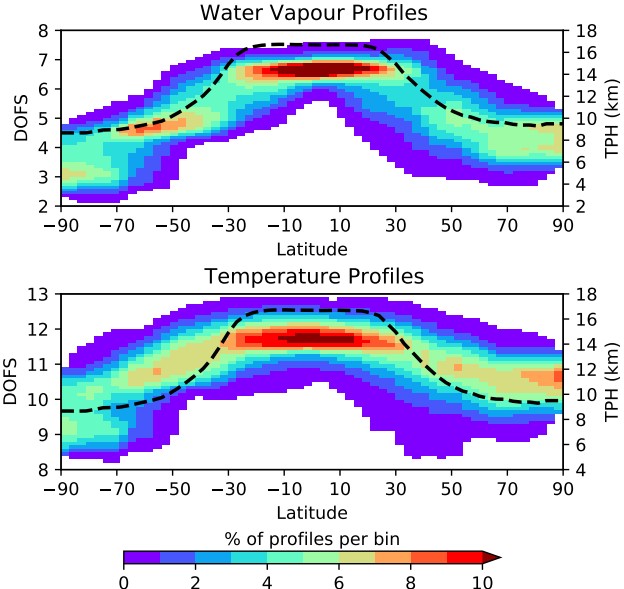

**Figure 3.** Visualisation of IMS water vapour and temperature profile degrees-of-freedom for signal (DOFS). This figure illustrates the latitudinal distribution of DOFS variability for both IMS water vapour and temperature profiles. DOFs were collected from the IMS L2 files between 2007-2016 and binned as a function of latitude. Values were then normalised using the total number of profiles in their respective latitude bin. The DOFS vary between 2-7 for water vapour and 8-13 for temperature, with strong peaks in the tropics. The spread in the data resembles the cold point tropopause height (TPH), especially for water vapour. The dashed black line represents the cold point TPH calculated from ERA5 temperature profiles (Hersbach et al., 2020).

Where the $\mathbf{G}$ is the gain matrix, and the subscript $f$ associated with the Jacobian matrix ($\mathbf{K}$) and averaging kernel ($\mathbf{A}$) denote that derivatives are computed with respect to perturbations on the fine atmospheric grid, $p_{atm}$, as opposed to the state vector. The 101 RTTOV pressure levels define this fine grid for the IMS algorithm. Therefore, the averaging kernel matrix is not square; rather, the two dimensions are the number of Eigenvector weights in the state vector and the 101 levels in the "true" profile. However, the (square) averaging kernel derived using the state vector weighting function (e.g $\mathbf{A} = \mathbf{G}\mathbf{K}$) would give the derivative of the retrieved Eigenvector weights with respect to true profile perturbations with shape given by the Eigenvectors. The practical uses of the square averaging kernel matrix ($\mathbf{A}$) are to i) smooth atmospheric profiles from models, reanalysis and *in situ* measurements (discussed further in Section 3.2), and ii) to obtain the degrees of freedom for signal (DOFS) for specific retrieval

products. The DOFS are given by the trace (sum of the diagonal elements) of the sub-matrix of $\mathbf{A}$ corresponding to a specific product and represent the total number of independent pieces of information in the profile. The averaging kernel of retrieved water vapour profiles (defined on the RTTOV levels) with respect to perturbations on the fine atmospheric grid is given by:

$$\mathbf{A}_{qf} = \mathbf{M}_{qf}\mathbf{A}_{f:q}, \tag{10}$$

where $\mathbf{A}_{f:q}$ is the averaging kernel for the water vapour state vector elements with respect to perturbations in ln(ppmv) on the fine atmospheric levels. The averaging kernel for temperature is derived similarly using the corresponding matrices. An understanding of a profile vertical resolution can be inferred from the DOFS value, as it describes the number of independent pieces of information resolved (Rodgers, 2000). Figure 3 shows the range of DOFS for IMS temperature and water vapour profiles as a function of latitude, with the tropopause height (TPH) overlaid. The two-dimensional (2D) histograms show the distribution of profile DOFS, from which we can see that in the tropics, most profiles sit between 6-7 and 11-12 DOFS for water vapour and temperature, respectively. Moving outwards through the mid-tropics to the high latitudes, the DOFS values reduce, with the distribution becoming more variable. Comparing the water vapour distribution to the cold point tropopause height (black dashed line), we can observe that they hold similar shapes, while for temperature, this is less so. This result is expected as nadir IR+MW sounders are predominately sensitive to the emissions from the troposphere, especially for water vapour.

The next conceptual step is how the DOFS relate to the vertical resolution of IMS profiles. Examples of averaging kernels for water vapour and temperature profiles from the IMS L2 product are given in Figure 4, where we see that most of the information for water vapour is situated in the lowest 10 km of the atmosphere while for temperature is more continuous into the lower stratosphere. Therefore, an examination of the cumulative degrees of freedom for signal (CDOFS) from these averaging kernels can be used to describe the vertical resolution of the retrieved profiles. The gradients of the CDOFS as a function of altitude can then be interpreted as the profile resolution at given heights. The desired performance for vertical resolutions from IASI is 1 & 2 km for temperature and water vapour profiles, respectively (Hilton et al., 2012). What can be seen from Figure 4 is that vertical resolution is not necessarily constant throughout the troposphere. Indeed, examining a different sounding over the same radiosonde site would show subtle differences. A key observation here is that the information from the water vapour profile terminates (vertical gradient) at the tropopause. Therefore, using the IMS water vapour profile above this height is meaningless.

## 2.2   Radiosonde Reference Measurements

This section outlines the two radiosonde records used as reference measurements in this study. The first source of radiosonde measurements used has been taken from the GCOS Reference Upper-Air Network (GRUAN) (Immler et al., 2010; Dirksen et al., 2014) archive, locations of the sites can be seen in Figure 5a. The scope of GRUAN is to provide long-term fiducial measurements, i.e. inclusion of uncertainty estimates) that can be used for calibration/validation exercises, studying atmospheric processes and determining trends. These high-resolution soundings are reported on time intervals of 2 seconds during the flight from the surface into the Upper Troposphere/Lower Stratosphere (UTLS) rather than the set pressure grid used by operational radiosonde archives. An advantage of the higher resolution of GRUAN measurements is that it captures changes in humidity gradients and temperature inversions which can be missed or underrepresented by standard and significant pressure levels. It should be noted that the soundings from GRUAN feature only the Vaisala RS92 radiosondes measurements and not the more recent (and accurate) RS41.

The second source of radiosonde data is taken from the Analyzed RadioSoundings Archive (ARSA). Produced at the Laboratoire de Météorologie Dynamique (LMD) since the late '90s, ARSA is designed for the processing and validation of level 1 (L1) and level 2 (L2) satellite data and applications. This includes forward and inverse radiative transfer simulations and inter-comparison of retrieved satellite geophysical parameters. The ARSA database is a global archive with observations from approximately 1450 stations. In the first instance, raw radiosondes observations with measurements between the surface and 300 hPa for water vapour and 30 hPa for temperature profiles are extracted from the ECMWF archive. These radiosonde observations are then extended above their highest measured point to 0.1 hPa with collocated data from ERA-Interim. Finally, level profile data from the SciSat Atmospheric Chemistry Experiment Fourier Transform Spectrometer (ACE-FTS) is used to complete the profile between 0.1 and 0.0026 hPa. The vertical resolution of ARSA varies within the profile, where the lowest part of the troposphere ranging from the surface to 800 hPa, has a resolution of 0.5 km. Between 800 and 200 hPa, the resolution is 0.8 km, increasing to 1.5 km from 200 hPa to 100 hPa. Above 100 hPa to the top-of-atmosphere (TOA), the resolution further reduces to 2.5 km. Unlike GRUAN, which applies several corrections to the raw measurement, e.g. correction to water vapour due to incident solar radiation on the radiosonde casing, rather the validation of every ARSA profile relies upon analysing the bias and standard deviation between observed satellite and simulated radiances (Scott, 2015). The ARSA measurement record started in January 1979 and is regularly updated on a monthly basis. Locations of 587 sites present in the archive during the study period

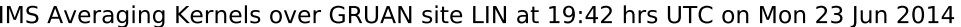

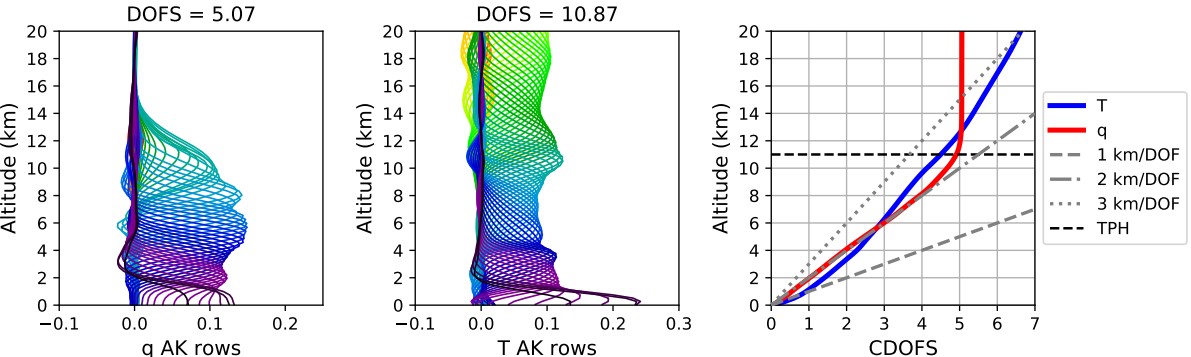

**Figure 4.** Example of averaging kernels (AK) for IMS water vapour (q) and temperature (T) profiles extracted over the GRUAN Lindenberg (LIN) site. The right-hand panel shows the cumulative degrees of freedom for signal (CDOFS), which illustrates how the vertical distribution information content can be related to profile vertical resolution. In the lower-mid troposphere, the water vapour CDOFS aligns with a vertical resolution of 2 km. Above 8 km, the gradient swiftly becomes zero around the tropopause height (TPH). The temperature profile starts at a 1 km vertical resolution and reduces to 2 km at 6 km. Above this height, the gradient shows the vertical resolution changes to ≈2.5 km, which remains consistent into the upper troposphere/lower stratosphere (UTLS) before further degrading to 3 km per degree of freedom (DOF).

can be seen in Figure 5b. For this study, we use the current version 2.7 archive, which has been in use since 2005.

Finally, it is worth noting that while radiosondes provide a source of reference data for profile validation, they are not without their own limitations and caveats of use:

- Model type: Corrections made to radiosondes are highly dependent on the make and model type, especially with older radiosondes (e.g. Miloshevich et al. (2001, 2006)). Both archives used in this study have different approaches to correct radiosondes, with GRUAN applying empirical corrections (Dirksen et al., 2014) and ARSA using a radiative transfer modelling to test for consistency between stable satellite radiances (Scott, 2015; Calbet et al., 2017).

- Time series consistency: Radiosonde archives are subject to semi-regular observation system changes, some of which are recorded by the WMO. For GRUAN, their certified sites undergo periodic auditing of their measurement programs and annual reviews to ensure all sites continue to meet practice standards. It is unclear how well this approach scales from ≈30 sites to 500+ found in a global network. ARSA uses the long-term statistics from the radiance intercomparisons to ensure quality consistency across the archive. This approach allows for a common method to be applied to a global network of up to 1450 sites; however, this relies on the radiometric stability of the reference satellite instrument.

- Sources of uncertainty: Radiosondes are subject to a number of sources of uncertainty which can be difficult to characterise fully. The GRUAN provides a comprehensive error budget for their products as their correction process allows estimates for each step. However,

ARSA, like other global datasets, does not give an uncertainty on the profiles it provides due to the complexity of such an exercise. In Trent et al. (2019), it was demonstrated that the uncertainty on operational records reduces to a few % ppmv with large collocation numbers.

- Distribution of sites: One of the strengths of operational radiosonde records is a large number of global sites available for match-ups. While ARSA does quality filter these, it still has over 500 sites within the study period. For GRUAN, there are only a small number of sites, though they try to sample major climate regimes to provide some global representation. A key weakness for any radiosonde archive is the lack of sites in the southern hemisphere, especially for GRUAN (Figure 5a).

## 3 Methodology

### 3.1 Collocation of IMS profiles at Radiosonde Sites

The framework for creating the satellite match-ups to ground truth used in this study has been developed within the ESA Water Vapour Climate Change Initiative (WV_cci) and builds from previous validations studies (Trent et al. (2018) & Trent et al. (2019)). Referred to from here on in as the Match-up Processor (MUP), this framework is designed to handle swath or gridded satellite data as well as several predefined *in situ* references. The match-up database (MUDB) is generated by supplying the MUP a driver file containing information on i) the dataset being validated, ii) the validation data record used as a reference, iii) what variable is being validated, (iv) the date range to process, and (v) which set of

## (a) GRUAN Sites (2007-2016)

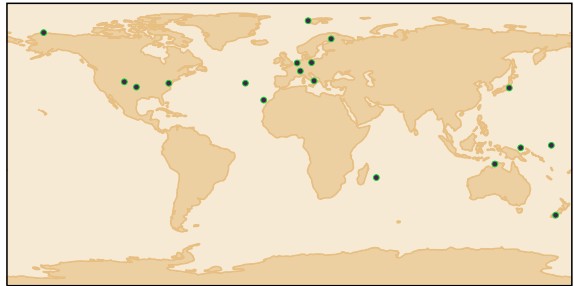

## (b) ARSA Sites (2007-2016)

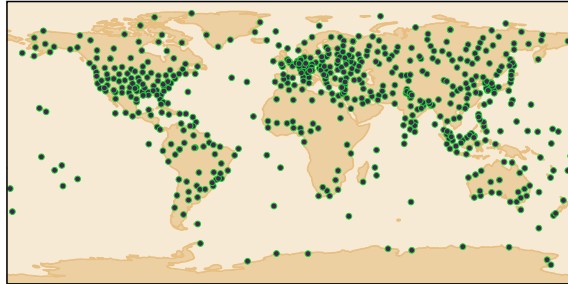

**Figure 5.** Locations of sites within the two radiosonde archives, a) the GCOS Reference Upper-Air Network (GRUAN), and b) the Analysed RadioSoundings Archive (ARSA). Locations are specific to upper-air soundings made between $1^{st}$ June 2016 and the $31^{st}$ December 2016. Further details regarding the individual GRUAN sites are given in Table A1.

collocation criteria to use. This approach allows for a flexible system that is capable of rapidly processing whole missions.

This study used broad criteria to maximise collocations for both radiosonde datasets. An IASI profile was initially considered collocated to a GRUAN or ARSA station if the satellite measurement fell within $\pm 3$ hr and 100 km of the radiosonde launch time. The IMS profile also required a consistent averaging kernel and uncertainty information to be propagated by the MUP. As the IMS scheme retrieves in cloudy and clear sky conditions, we accept all scenes with up to 80% cloud cover (Susskind et al., 2006). Finally, an additional quality filter was applied to all matched cases that fell within these criteria to reduce uncertainty. Levels within all profiles were excluded if the IMS water vapour profile uncertainty was above 50% ppmv. This scenario predominately was found to occur for IMS profiles only at high altitudes in the troposphere, resulting in lower density sampling, which will introduce some noise to the analysis. However, this is minimised by calculating global or per latitude band statistics which uses large numbers of matched pairs, unlike for

site comparisons with a low number of matched cases. When using broad collocation criteria, any mismatch introduced during the match-up will affect the performance of individual comparison performance (Sun et al. (2010) & Sun et al. (2017)). Therefore, a robust statistics approach was adopted to minimise this effect as demonstrated in Trent et al. (2019).

### 3.2 Comparison of IMS profiles with Radiosonde

Retrieved temperature and humidity profiles from IASI, MHS & AMSU-A represent the best estimates of the atmospheric state, to which a smoothing function has been applied (Rodgers and Connor, 2003). Therefore, averaging kernels from the IMS L2 product are used to smooth (or convolve) the radiosonde profile to the vertical resolution of IASI. This allows for like-for-like comparisons between the retrieved and reference profile. For radiosonde temperature profiles, the averaging kernel is applied thus:

$$\mathbf{x}_{est} = \mathbf{x}_o + \tilde{\mathbf{A}}(\mathbf{x}_t - \mathbf{x}_o), \tag{11}$$

where $\mathbf{x}_o$ is the IMS a priori profile, $\tilde{\mathbf{A}}$ is the averaging kernel that has been reconstructed onto the 101 level retrieval grid, $\mathbf{x}_t$ is the radiosonde reference profile on the 101 level grid, and $\mathbf{x}_{est}$ is the convolved reference profile. In the thermal infrared (TIR), changes in column density of water vapour have greater linearity in log space relative to any absolute change. Therefore, for humidity profile comparisons, equation 11 is rewritten as (Maddy and Barnet, 2008):

$$\ln(\mathbf{x}_{est}) = \ln(\mathbf{x}_o) + \tilde{\mathbf{A}} \times \ln\left(\frac{\mathbf{x}_t}{\mathbf{x}_o}\right). \tag{12}$$

Next, values are calculated for weighted layers ($\mathbf{x}_{(z)}$) within each profile, where the layer boundaries are defined by standard pressure levels defined at 1000, 925, 850, 700, 500, 400 and 300 hPa:

$$\mathbf{x}_{(z)} = \frac{\sum_{l=1}^{n} \mathbf{x}_{(l)} \mathbf{P}_{(l)}}{\sum_{l=1}^{n} \mathbf{P}_{(l)}}. \tag{13}$$

Where $\mathbf{x}_{(l)}$ is the convolved radiosonde or IMS profile value at level $l$, $\mathbf{p}_{(l)}$ is the pressure profile value at level $l$, and $n$ is the numbers of levels in the layer. Weighted layer mean profiles are not calculated for altitudes higher than 300 hPa because ARSA profile values are taken from ERA-Interim in the upper troposphere/stratosphere. All statistics used in this study are calculated from the layer mean profiles. Firstly, for each layer, we calculate the systematic difference or bias ($b_{(z)}$). As with Trent et al. (2019), we use the median difference:

$$b_{(z)} = median(\mathbf{x}_{(z)} - \mathbf{x}_{est(z)}), \tag{14}$$

where $\mathbf{x}_{(z)}$ and $\mathbf{x}_{est(z)}$ are the profile values for layer $z$ for IMS and the radiosonde respectively. Water vapour profile values will vary by up to four orders of magnitude between the surface and upper troposphere. Therefore, the layer bias is normalised by the median radiosonde layer value ($\bar{\mathbf{x}}_{est(z)}$):

$$\hat{b}_{(z)} = \frac{b_{(z)}}{\bar{\mathbf{x}}_{est(z)}}. \tag{15}$$

Profiles from GRUAN, unlike ARSA, are provided with estimates of the uncertainty for each measurement. These can then be propagated to provide corresponding uncertainties for profile measurements. However, when averaging over large numbers of collocations, the uncertainty on the bias reduces below 1 % ppmv. In the Trent et al. (2019) study, bias uncertainties for AIRS were shown to reduce to between 0.15-0.43 % ppmv for global matches to GRUAN. Although, what is difficult to calculate accurately and is not accounted for in tropospheric profile validation studies is the collocation uncertainty. While the collocation uncertainty will also reduce with averaging large numbers of matches, broad collocation criteria and atmospheric variability mean this uncertainty will still dominate the total error budget. Therefore, we can think of the variability of the median as an estimate of the precision of the bias. To quantify the spread about the median, we calculate the median absolute deviation ($\sigma_{(z)}$), a robust measure of the data variability:

$$\sigma_{(z)} = median|(\mathbf{x}_{(z)} - \mathbf{x}_{est(z)}) - b_{(z)}|. \tag{16}$$

As we use robust statistics, the median absolute deviation (MAD) values cannot be treated in the same way as standard deviation and used to calculate the standard error by dividing through by $\sqrt{N}$. For water vapour, MAD values are also normalised by the median radiosonde layer value:

$$\hat{\sigma}_{(z)} = \frac{\sigma_{(z)}}{\bar{\mathbf{x}}_{est(z)}}, \tag{17}$$

where $\hat{\sigma}_{(z)}$ is the normalised layer MAD. Scaling the normalised values by $10^2$ presents the units for both the bias and median absolute deviation in % ppmv. This allows biases at different layers to be relatable. For future studies, the approach for handling collocation uncertainty can be made more sophisticated than is outlined here. A new study from Laeng et al. (2022) provides a framework to account for the natural variability of atmospheric mixing ratios, allowing for such estimates. At the time the work of our study was undertaken, this tool was not available and as such not discussed further.

Finally, for examining the stability in the observed biases, a level shift regression model is used to calculate the trend in the monthly IMS layer (Weatherhead et al. (1998), Mieruch et al. (2014)):

$$Y_t = \mu + \omega X_t + \delta U_t + \eta_t, \qquad t = 1, 2, 3, \ldots, N, \tag{18}$$

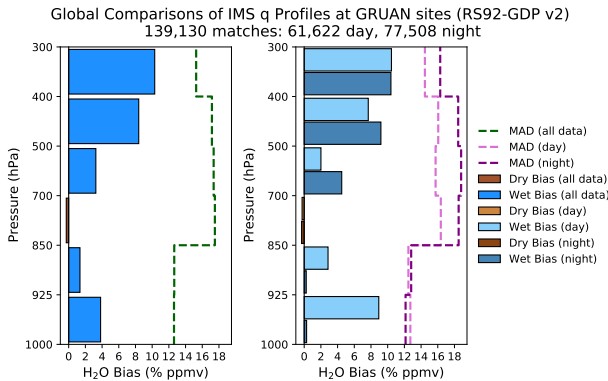

**Figure 6.** Comparisons of water vapour profiles from the IMS L2 product at GRUAN sites, matched at sites between June $1^{st}$ 2007 and December $31^{st}$ 2016 and with up to 80% cloud cover. Median biases for atmospheric layers are shown with blue bars representing a median wet bias relative to GRUAN, while brown bars depict a median dry bias with respect to GRUAN. The left-hand plot shows results for all data that passes quality control, with the right-hand plot showing the breakdown between day and night results. All biases have been normalised by the median GRUAN layer value for the site and multiplied by 100, i.e. to scale to % ppmv. Dashed lines represent each layer's normalised Median Absolute deviation (MAD).

where $Y_t$ is the bias at time t, $\mu$ is the intercept, $\omega$ is the trend in the bias, $X_t$ is the time index, $\delta$ is the magnitude of any shift, $U_t$ is the step function, and $\eta_t$ is the fit residual. For this study, the step function is assumed to be negligible (i.e. $\delta U_t = 0$) because the IASI instrument is considered to be a stable reference. This is evidenced by the use of IASI brightness temperatures for calibration by the Global Space-based Inter-Calibration System (GSICS) for other infrared (IR) satellite sensors (Goldberg et al., 2011). For the residuals, the same approach is used from Schröder et al. (2019) where four frequencies (asymmetric fitting of the annual cycle) and ENSO strength are fitted simultaneously.

## 4   Validation over GRUAN sites

### 4.1   Water Vapour Biases

Results were computed for matches made in cloudy-sky conditions (up to 80% cloud covered) over 17 of the 18 GRUAN sites (see table A1 for the complete list), with Darwin being the only site with no cases found. Matches were further subdivided into day and night scenes using the solar zenith angle from the IMS L2 product. Figure 6 presents the results for IMS water vapour profile biases, median absolute deviations for all scenes, and the split between day-night cases. IMS biases show a generally low wet bias relative to GRUAN, which increases with altitude. The lowest bias is found in the mid-tropospheric layer between 850-700 hPa, where a slight

**Table 1.** Breakdown of GRUAN site specific IMS water vapour profile biases and standard errors, given in % ppmv for the same layers shown in Figure 6. The site short names correspond to the details given in Table A1 and are accompanied by the total number of matches made with IMS. Sites appear in latitudinal order (north to south). After quality filtering, $\approx 90\%$ of all matches remain at all levels.

| Site (Nmatches) | 1000-925 hPa | 925-850 hPa | 850-700 hPa | 700-500 hPa | 500-400 hPa | 400-300 hPa |
|---|---|---|---|---|---|---|
| BAR (7389) | 9.99±15.61 | 10.32±15.33 | 1.27±16.30 | 5.01±17.95 | 21.53±16.74 | 27.63±18.58 |
| NYA (7919) | 19.01±16.90 | 17.49±15.43 | 11.29±17.52 | 4.05±16.46 | 12.77±15.65 | 12.57±18.33 |
| SOD (2225) | 11.64±14.08 | 7.23±12.61 | -1.50±14.50 | 3.15±16.28 | 14.03±16.25 | 17.17±17.13 |
| LIN (100654) | 2.54±12.21 | -0.41±12.14 | -0.72±17.52 | 3.04±17.45 | 7.45±17.72 | 9.66±15.25 |
| CAB (843) | 8.81±14.79 | 6.08±14.68 | 2.28±20.18 | 3.14±20.34 | 12.02±20.08 | 12.27±17.38 |
| PAY (122) | 12.46±17.67 | 11.06±16.92 | 6.70±18.59 | 6.38±16.21 | 12.27±16.20 | 5.56±16.18 |
| POT (126) | 4.60±12.27 | 10.48±16.59 | -0.14±15.37 | 2.78±17.99 | 5.42±18.89 | 7.92±11.34 |
| BOU (161) | —— | —— | 10.26±19.57 | 1.18±10.40 | 5.94±15.30 | 3.85±12.43 |
| GRA (48) | 9.92±8.30 | 1.26±6.67 | -9.75±11.11 | -6.53±12.92 | 3.49±15.40 | 18.12±14.59 |
| BEL (34) | 20.11±19.47 | 14.27±26.85 | 11.07±27.28 | 16.38±25.25 | 11.91±19.09 | 9.19±13.80 |
| SGP (4861) | 4.67±11.89 | 2.10±10.53 | -4.26±13.38 | -0.83±13.42 | 5.60±15.73 | 9.90±15.01 |
| TAT (12369) | 4.70±13.65 | 2.17±14.30 | -4.04±19.59 | 8.29±21.87 | 12.65±20.18 | 12.35±17.69 |
| TEN (1808) | 11.21±10.99 | 5.29±14.01 | -3.74±19.62 | 4.13±17.18 | 3.49±16.14 | 8.60±14.94 |
| NAU (298) | -1.54±5.45 | -0.22±5.47 | -5.03±7.78 | -7.65±9.84 | 0.80±14.40 | 1.55±12.51 |
| MAN (111) | -2.55±5.62 | -4.59±4.28 | -4.17±8.81 | -3.85±7.17 | -12.31±13.78 | 2.79±11.35 |
| REU (18) | —— | —— | 14.53±22.19 | -3.08±10.03 | 0.22±10.36 | 6.39±9.44 |
| LAU (144) | 18.86±21.26 | 15.98±22.03 | 6.41±21.48 | 4.62±18.08 | 12.99±15.78 | 15.71±15.62 |

—— == layers below surface pressure or do not contain full profile information.

dry bias of -0.29±17.51 % ppmv is observed. This layer coincides with the overlap in peak vertical sensitivities of the IASI 6 $\mu$m region and MHS 183 GHz channels. In the lower troposphere (1000-850 hPa), daytime biases dominate with a high of 8.93±12.72 % ppmv seen in the surface layer. The inverse is valid for the mid-to-upper tropospheric layers (700-400 hPa), where nigh-time biases are larger than the equivalent daytime biases by about 2 % ppmv in both layers. The upper-tropospheric layer (400-300 hPa) displays a consistent wet bias across the day and night scenes (10.45±14.46 to 10.39±16.29 % ppmv respectively).

Table 1 presents a breakdown of individual site biases, median absolute deviations, and the number of collocations for each GRUAN site between June 2007 and December 2016. The first point is that sampling differences can be up to three orders of magnitude because GRUAN does not have regular launch data for each site. The Lindenberg (LIN) lead site provides approximately 72% of all matches made to GRUAN radiosondes. Therefore, 'global' biases are weighted towards the Lindenberg site result. GRUAN sites situated at high latitudes in the northern hemisphere tend to show wetter biases for the lower-to-mid tropospheric layers. In contrast, GRUAN sites in the Tropical Warm Pool (TWP) see persistent dry biases below 500hPa. Northern hemisphere high-latitude sites also have a lower performance in the upper troposphere layer 400-300 hPa, with Barrow (BAR) seeing a wet bias of 27.63±18.58 % ppmv.

## 4.2 Temperature Biases

The same exercise is repeated for IMS temperature profiles, with biases reported in Kelvin (K). Figure 7 shows the results for biases calculated for all matches over all sites and the diurnal split between day and night cases. IMS temperature biases are within ±0.2 K for the first scenario. The bottom and top layers both see cold biases of -0.18±0.49 and -0.17±1.16 K, respectively, while the rest of the troposphere shows warm biases between 0.06±0.62 and 0.21±0.55 K. As with water vapour, temperature biases in the lowest layer (925-1000 hPa) are dominated by the daytime bias (-0.44±0.99 K). This negative daytime bias continues up to 500 hPa, with the magnitude reducing with altitude. Night-time biases below 400 hPa show warm biases between 0.11±1.31 and 0.32±1.07 K, with the largest seen in the mid-tropospheric layers between 700 to 925 hPa. The night-time bias dominates the mid-tropospheric temperature bias for all sites and all matches. The median absolute deviation ranges between 0.43 to 1.31 K for scenarios with a decrease in magnitude consistently observed as a function of altitude. Higher variability is observed for night-time temperature biases relative to the daytime, mirroring the behaviour seen for water vapour.

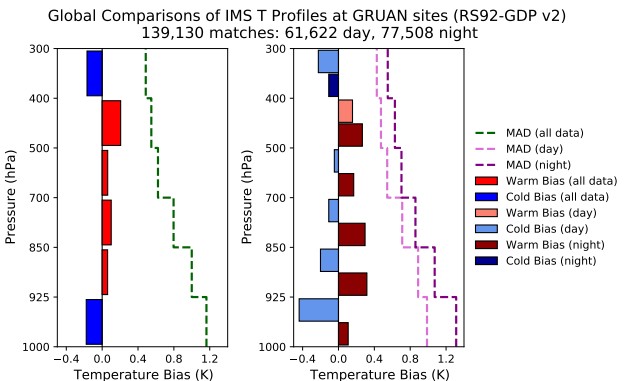

**Figure 7.** As Figure 6, but for comparisons for IMS IASI temperature profiles at GRUAN sites. Median biases for atmospheric layers are shown with red bars representing a median warm bias relative to GRUAN. In contrast, dark blue bars depict a median cold bias with respect to GRUAN. The dashed lines represent the layer Median Absolute Deviation (MAD) in Kelvin.

The breakdown of IMS temperature profile biases by GRUAN site is given in Table 2. As expected, temperature biases for all sites and matches are weighted towards Lindenberg (LIN) results. The surface layer negative bias is a common feature for stations situated in the mid-latitudes and tropics, with the coldest bias value of -1.57±0.7 K seen over Manus (MAN). However, the site at Potenza, Italy (POT) displays a different behaviour with a warm bias of 2.44±1.7 K in the surface layer. With only 126 matches over the whole period, this will have little impact on the collectively observed bias. A majority of sites see a cold bias in the mid-troposphere (400-925 hPa), whereas the 'all sites/all matches' result (Figure 7) shows a small warm bias for these layers. However, the sites which exhibit the warm bias also tend to have a higher number of matches (>1000) and are

**Table 2.** As Table 1, breakdown of GRUAN site specific IMS temperature profile biases and standard errors, given in K for the same layers shown in Figure 7.

| Site (Nmatches) | 1000-925 hPa | 925-850 hPa | 850-700 hPa | 700-500 hPa | 500-400 hPa | 400-300 hPa |
|---|---|---|---|---|---|---|
| BAR (7389) | 0.06±1.23 | 0.75±0.96 | 0.75±0.83 | 0.69±0.63 | 0.68±0.55 | -0.13±0.45 |
| NYA (7919) | 0.87±1.40 | 0.93±1.01 | 0.69±0.89 | 0.44±0.58 | 0.57±0.48 | -0.06±0.43 |
| SOD (2225) | 0.01±1.04 | 0.30±0.96 | 0.40±0.75 | 0.34±0.60 | 0.38±0.56 | -0.23±0.47 |
| LIN (100654) | -0.19±1.11 | 0.00±0.98 | 0.04±0.76 | 0.00±0.60 | 0.18±0.53 | -0.19±0.48 |
| CAB (843) | -0.30±1.31 | 0.00±1.24 | -0.21±0.97 | -0.16±0.77 | 0.04±0.74 | -0.35±0.67 |
| PAY (122) | -0.39±1.80 | -0.31±1.69 | -0.21±1.24 | -0.20±0.61 | 0.15±0.51 | -0.25±0.47 |
| POT (126) | 2.44±1.70 | 1.58±1.10 | 0.30±0.62 | -0.20±0.45 | 0.01±0.44 | -0.16±0.41 |
| BOU (161) | —— | —— | -1.23±1.06 | -0.95±0.64 | -0.02±0.49 | -0.02±0.49 |
| GRA (48) | -0.43±0.66 | -0.22±0.46 | -0.48±0.61 | -0.45±0.50 | -0.14±0.55 | -0.21±0.33 |
| BEL (34) | -1.75±0.99 | -1.01±1.08 | -0.40±1.09 | 0.03±1.05 | -0.10±0.50 | -0.43±0.40 |
| SGP (4861) | -0.20±1.27 | -0.25±0.89 | -0.19±0.73 | -0.17±0.54 | 0.02±0.46 | -0.08±0.44 |
| TAT (12369) | -0.74±1.24 | -0.29±0.96 | -0.04±0.85 | 0.01±0.66 | -0.08±0.61 | -0.16±0.62 |
| TEN (1808) | -0.55±1.45 | 0.08±1.26 | 0.43±1.03 | 0.10±0.66 | 0.07±0.53 | 0.01±0.51 |
| NAU (298) | -0.90±0.64 | -0.11±0.49 | -0.26±0.40 | -0.43±0.37 | -0.12±0.32 | -0.06±0.34 |
| MAN (111) | -1.57±0.70 | -0.55±0.46 | -0.18±0.44 | -0.58±0.32 | -0.34±0.32 | -0.12±0.29 |
| REU (18) | —— | —— | 0.20±0.71 | -0.37±0.47 | -0.34±0.48 | -0.02±0.36 |
| LAU (144) | -0.90±1.30 | -0.72±1.16 | -0.36±0.74 | -0.17±0.60 | 0.08±0.42 | -0.42±0.48 |

—— == layers below surface pressure or do not contain full profile information.

mainly found at higher latitudes, e.g. Barrow (BAR), Lindenberg (LIN), Ny-Ålesund (NYA), and Sodankylä (SOD).

### 4.3 Biases Dependence on Cloud Fraction

A key benefit of using the combination of IR and MW instruments for NWP is the ability to produce water vapour and temperature profiles in clear and cloudy scenes. However, it has been shown for the Atmospheric Infrared Sounder (AIRS) that cloud amount, and type can impact profile biases (Hearty et al., 2014; Wong et al., 2015; Trent et al., 2019).

Therefore, understanding the impact of cloud fraction within the IASI IFOV on IMS profile biases is also of interest to this study. IMS profile biases were binned according to cloud fraction at intervals of 0.1 for all sites for all matches (day & night cases) and for the separate day and night cases. Water vapour and temperature bias results as a function of cloud fraction are presented in Figure 8 along with the difference between day and night cases to the 'all cases' result. It should be noted that a BTD flag is used to remove cloudy scenes that significantly impact the retrieval. While IMS can produce profiles for cloudy IFOVs, the BTD flag will disproportionately remove some of the profiles across increasing cloud fractions. This explains the distribution we observe in Figure 8.

Water vapour profile biases shown in Figure 8 indicate that the wet bias seen in the lowest tropospheric layer (925-1000 hPa) is weighted towards clear-skies or scenes with a cloud fraction below 0.1 (or 10%). The stronger wet bias observed in the upper tropospheric layers (300-500 hPa) is more sensitive to cloud amounts > 10%. At higher cloud fractions, the wet bias is seen to double observed <10% cloud cover to 18.87 % ppmv. Cloud amount can also be attributed to the slight warm biases observed in the mid-to-upper troposphere (925-400 hPa) relative to GRUAN. The most affected layer is

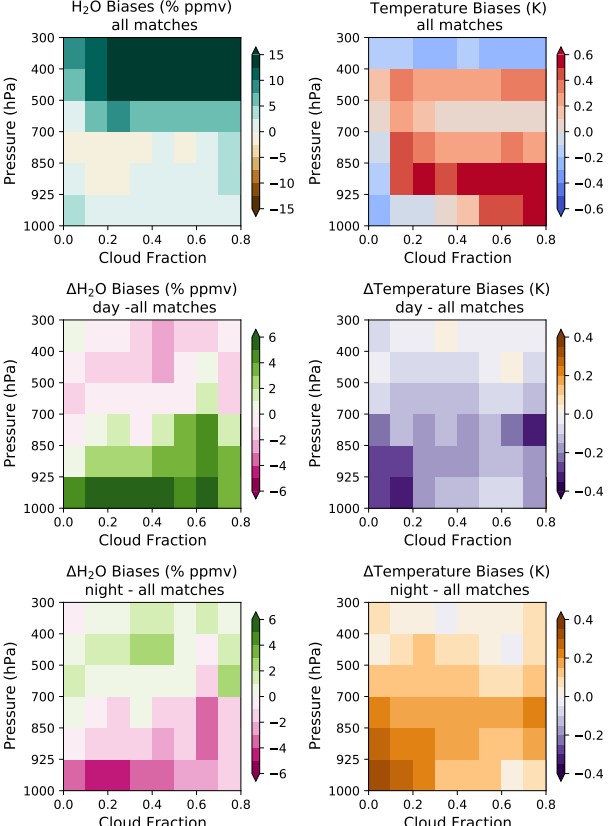

IMS profile biases as a function of cloud fraction at GRUAN sites

**Figure 8.** IMS water vapour (left-hand-side) and temperature (right-hand-side) profile biases as a function of cloud fraction for all sites, calculated for 10% cloud faction bins for each tropospheric pressure layer. Differences between daytime and night-time to the all-sites bias are shown for water vapour and temperature, respectively.

found between 925-850 hPa, with the maximum warm bias of 0.73 K above 50% cloud fraction. For the 500-400 hPa layer, the biases seen above 60% cloud fraction dominate the result seen in 7. Whereas, for the 925-850 hPa layer where the strongest biases are found, biases seen in the global all-site result are being significantly weighted by cold biases seen in cloud fractions < 10%.

The split of matches into daytime and night-time cases also reveals a diurnal dependence of the cloud fraction and observed biases. From visual inspection, an apparent 1:1 gradient (running bottom left to top right) is observed in both scenarios, splitting behaviour seen in day and night cases relative to the cumulative result. Daytime biases are up to 3.88% drier relative to those seen for the global all-site result above the 1:1 split, while below it, wetter biases are seen for daytime matches with larger differences of 5.08% to 6.91% observed. The inverse of this relationship is seen for night-time results. The region above the 1:1 gradient is wet-biased by up to 2.29% relative to all matches and dry-biased below the

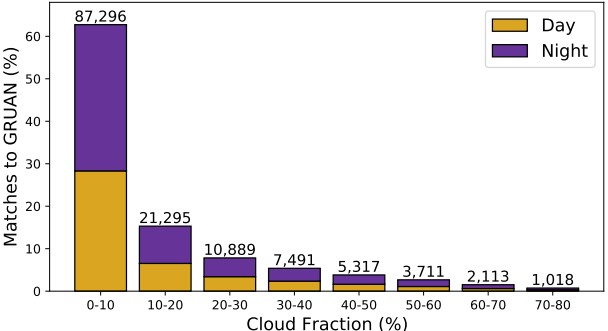

**Figure 9.** Stacked histogram of IMS collocations to GRUAN sites split into 10% cloud cover intervals. Results are separated into day and night cases, with the total number of matches shown at the top of each bar. Further subdivision of the first bin (0-10%) can be seen in Figure B1 of the Appendix.

1:1 division. Biases observed in this region again see larger differences in all matches for night-time cases, with a maximum difference of -4.45% seen in the lowest tropospheric layer (1000-925 hPa).

The diurnal pattern for temperature biases displays a more monotonic behaviour than water vapour. Daytime biases are almost exclusively colder than the all matches result, while night-time biases are more warm-biased. In both scenarios, the lower troposphere (below 850 hPa) with cases up to 20%
cloud cover shows the greatest differences of -0.28 K and 0.3 K for day and night matches, respectively.

   Figure 9 illustrates sampling across the cloud fraction bins in absolute and relative terms. Over 60% of all matches are found for scenes with 0-10% cloud cover, with 40% of all those cases found to be when the IMS IFOV is 0-1%
(e.g. clear-skies). The number of matched pairs drops off significantly with increasing cloud fraction. For the highest cloud cover category (70-80%), only 0.7% of cases remain. While sampling of cloud cover reduces in frequency as cloud fraction increases, relative sampling between day and night
scenes for each bin is reasonably consistent with an average 43%/57% split, respectively.

## 5   Validation over ARSA sites

While radiosonde archives such as ARSA do not contain the
25 same level of fiducial information as GRUAN, a key advantage is greater global sampling. A multi-year time series can yield match-up numbers 1 to 2 orders of magnitude greater than with a smaller network like GRUAN. Therefore, analysis against GRUAN expands on the global results by splitting
match-ups into five latitudinal bands. Finally, global and latitudinal bias trends are examined to assess the stability relative to Global Climate Observing System (GCOS) requirements.

### 5.1   Global Profile Biases

Collocation of IASI soundings against ARSA radiosonde
measurements between $1^{st}$ June 2007 and the $31^{st}$ of December 2016 yields over $1.2 \times 10^6$ matched pairs for analysis, with a 59% and 41% split between day and night overpasses respectively. Water vapour profiles are wet-biased between 0.38 to 6.54 % ppmv relative to ARSA, with the larger bi-
ases seen in the 1000-925 hPa and 500-400 hPa layers (Figure 10a). Like comparisons to GRUAN, the smallest bias is seen in the mid-troposphere. The spread of biases measured by the median absolute deviation shows the same behaviour as GRUAN results, with values ranging from 11.85 to 18.35
45 % ppmv and the larger values occurring between 850-400 hPa. Daytime cases dominate the observed wet bias at each layer seen in the all matches result, with a maximum of 9.39 % ppmv seen in the lowest tropospheric layer. In contrast, night-time biases drop below 4 % ppmv, with all layers show-
50 ing lower biases than the 'all matches' and daytime results. In the mid-troposphere, the night-time is again the smallest in magnitude, though it switches from a wet to a dry bias (Figure 10b).

   Temperature profile biases are found to be within -0.39 K
to 0.06 K relative to ARSA with a predominant cold bias (Figure 10c). The observed biases' variation is highest in the surface layer, with a median absolute deviation of 1.13 K. The magnitude of the layer median absolute deviations reduces in altitude, with a value of 0.46 K seen in the upper
tropospheric layer (400-300 hPa). Figure 10d highlights that daytime matches dominate IMS profile cold bias seen for all matches, while night-time matches exhibit a small warm bias (0.07 K to 0.22 K) between 925-400 hPa. With more than $2 \times 10^5$ more matched pairs, daytime median absolute devi-
ation values are constantly lower than those calculated for night-time collocations. However, these differences are less than 0.12 K on average.

   Figures 11a-c show the impact of cloud fraction on the IMS water vapour profiles biases on all, day and night
matches, respectively. In general, increasing cloud amount slightly reduces the wet bias relative to clear skies below 850 hPa while increasing the wet bias between 850-400 hPa. This pattern is also observed for daytime collocations, though biases are wetter by up to 4.5 % ppmv than those seen
for all matches. Similarly, night-time collocations show the same behaviour, except biases tend to be drier relative to all matches by as much as -3.35 % ppmv.

   Figures 11d-f, for all day and night cases, respectively, the same results for temperature biases. The cloud faction im-
80 pact on all matches shows an average warm bias of 0.2 K, with a maximum of 0.44 K for cloud fractions above 10%. In the upper tropospheric layer (400-300 hPa), the cold bias increase in magnitude by ≈0.1 K with increasing cloud faction within the field of View (FOV). Separation of the diurnal
effects of cloud fraction on temperature profile biases has the same behaviour seen over GRUAN sites. Daytime biases are

Global and Latitudinal Band Comparisons of IMS q and T Profiles at ARSA sites

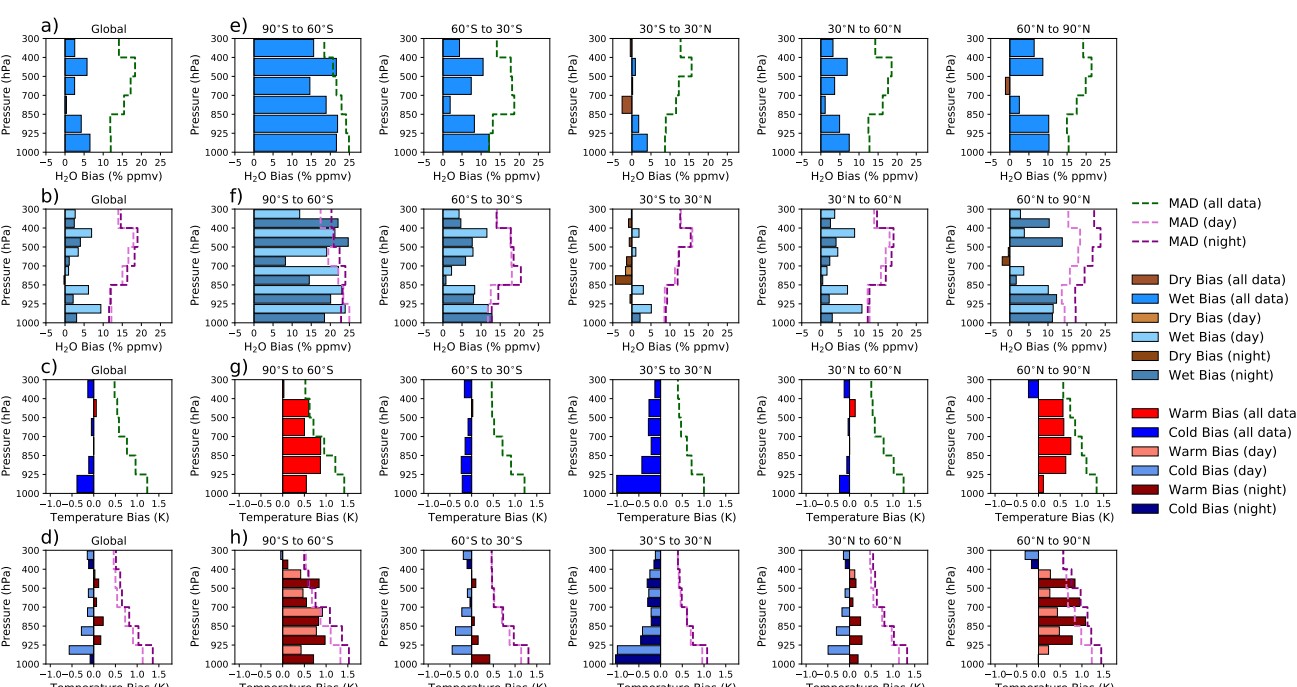

**Figure 10.** IMS profile biases relative to ARSA radiosonde measurements. Profile biases are shown for all global water vapour (a) and temperature (c) results as well as the split between day and night biases (b & d respectively) in the same manner as Figures 6&7. Biases for five broad latitude bands are also shown for water vapour and temperature comparisons (e & g). Day/night biases for each latitude band are also given in f & g for water vapour and temperature profiles, respectively. Median absolute deviations (MAD) are shown as dashed lines for all matches and the day and night split.

colder relative to the all-site result by up to -0.37 K, while night-time biases are warmer by as much as 0.4 K. The larger differences in both cases are seen below 700 hPa, nearer to the surface and for cloud fractions greater than 30%.

## 5.2 Latitudinal Dependence on Biases

To investigate how biases change with latitude, collocations are binned into five broad bands that span 90°S-60°S, 60°S-30°S, 30°S-30°N, 30°N-60°N, and 60°N-90°N. Due to the disproportionate distribution of global radiosonde sites (see Figure 5b), the percentage of match-ups are split 0.8%, 4.7%, 21.7%, 62.9%, and 9.9% between the bands respectively. Once separated, matches were processed in the same manner as global results to produce biases for all, day, and night cases and cloud fraction dependence.

Figure 10e shows IMS water vapour profile biases per latitude band for all matches. The largest biases are observed between 90°S-60°S, where values are >20 % ppmv in most layers. However, there are only ten sites along the Antarctic coastline at this latitude. The mid-latitude bands show similar performance with wet biases between 1-12 % ppmv, with the largest biases seen in the 1000-925 hPa surface layer. The northern mid-latitude band is slighter and higher perform-

ing with wet biases 3% ppmv lower on average. Biases in the tropical band are the lowest ranging between -2.6 and 4 % ppmv below 700 hPa and -0.4 and 0.9 % ppmv above 700 hPa. Finally, the Artic band (60°N-90°N) sees predominately wet biases below 10 % ppmv with a small 1 % ppmv dry bias seen between 700-500 hPa. All bands show the same median absolute deviation distributions with varying magnitude except for the Antarctic band, where the highest values are seen at the surface (25 % ppmv), reducing with altitude. The daytime wet bias dominance observed in the global all result (Figure 10b) is also seen in the northern mid-latitude band. This is not coincidental, as 60% of all daytime cases are found between these latitudes. The other latitude bands show differing patterns, though most of all biases are wet relative to ARSA. The main exception to this behaviour is in the tropics, where night-time values above 850 hPa (free troposphere) are dry-biased up to 4.3 % ppmv. Examination of the impact of cloud fraction shows Northern Hampshire mid-latitudes strongly influence what is observed for all global results (Figure 11g). At other latitudes, comparisons to ARSA generally present wetter biases except for the Tropics. Here we observe the (overall) lowest wet biases and a persistent dry bias between 850-700 hPa. The strong wet biases are seen below 60°S in Figure 10e continue across all cloud amounts.

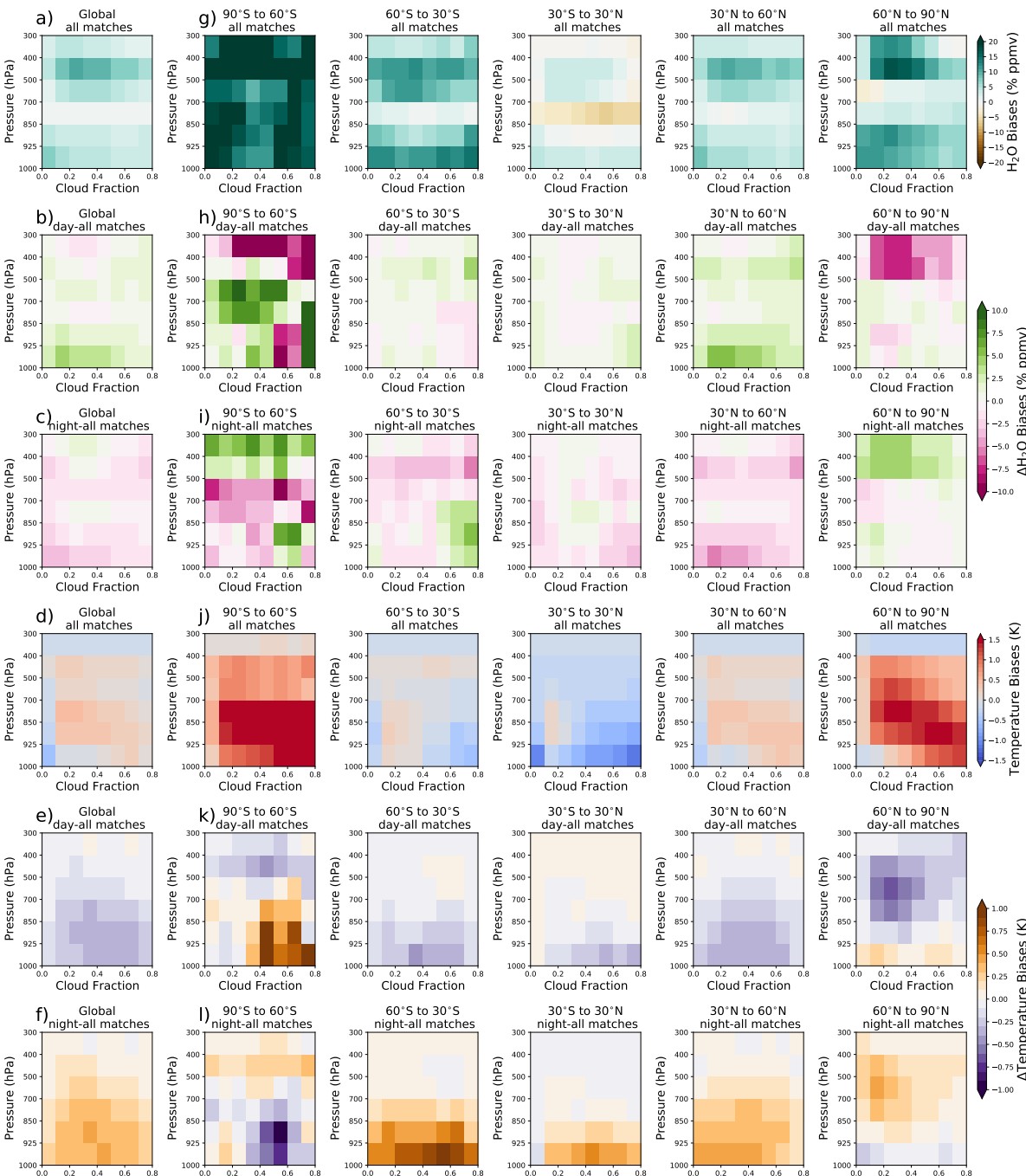

**Figure 11.** Cloud fraction effects on IMS biases relative to ARSA radiosonde water vapour and temperature profiles. Results here are broken down into; global water vapour and temperature biases (a, d), daytime differences to global biases (b, e), night-time water vapour and temperature bias differences (c, f), and biases calculated for five broad latitudinal bands (g, j) and the day and night-time differences (h, i & k, l respectively).

It is worth noting that strong wet biases can be correlated with low sampling, especially in low and upper tropospheric layers. Diurnal behaviour seen for changes in cloud fraction in the global comparisons to ARSA prevails when split in latitude bands with one key difference. At high latitudes above 500 hPa, biases show opposing results. When daytime biases

are normally wet, they become dry-biased relative to global results. For night-time, the inverse is observed where the expectation is that biases are drier relative to the global result and are now wetter. This effect could arise from upper tropospheric layer sensitivity to the tropopause and lower stratosphere, similar to what has been observed for AIRS (Trent et al., 2019).

Results for IMS temperature profile comparisons to ARSA radiosondes for different latitude bands and the split between day and night match-ups are shown in Figures 10g&h respectively. IMS shows a dominant cold bias at tropical and mid-latitudes relative to ARSA. Here values range from -1.01 K in the surface layer in the tropics to 0.13 K in the mid-to-upper troposphere between 30°N-60°N, with an average bias of -0.2 K. Whereas, at high latitudes, a warm bias is observed with maximum values between 0.74-0.87 K. However, unlike in lower latitudes, polar biases peak throughout the free troposphere rather than the surface layers. Analysis of diurnal biases shows that outside tropical latitudes, night match-ups are consistently warm biased, with a strong dominance above 60°N. Mid-latitude daytime biases are cold relative to ARSA temperature profiles, where the distributions and magnitudes negate the warm biases in the 'all' match-up results. In the tropics, day and night-time biases are similar, with the magnitude of the night-time results only 0.03 K colder on average relative to daytime biases. The spread of biases exhibits a high-level consistency across the latitudinal bands relative to both one another and the global results. Median absolute deviation values range from 0.96-1.3 K near the surface, degrading to 0.4-0.57 K in the upper troposphere. Like the global results, night-time variability is higher than daytime matches, reaching a maximum of 0.26 K in the lower troposphere.

Temperature biases as a function of cloud are shown in Figures 11j-l for all, day and night differences, respectively. Like water vapour, temperature biases for the 30°-60∘ band show a strong resemblance to the global all matches result (Figure 11d). The significant influence of cloud fraction on temperature profile biases is observed at high latitudes. Polar temperature biases can become increasingly warm-biased as the cloud amount within the IFOV increases. There is also a vertical dependence on this behaviour as the sensitivity to cloud fraction reduces with altitude, i.e. the surface layer biases respond continuously to cloud fraction, while other altitudes peak at lower cloud amounts. Temperature biases for the surface layers compared to ARSA soundings reach 1.5 & 3.3 K for Northern and Southern polar sites, respectively. Diurnal patterns generally follow the global results with warmer night-time biases compared to daytime collocations. However, there are a few occasions where this behaviour is flipped. Most notable are the matches between 90°S and 60°S, where daytime near-surface layers show biases peak differences 1.36-1.77 K warmer relative to night-time biases for cloud fractions above 40%. The second region is the Northern polar region where the surface tropo-

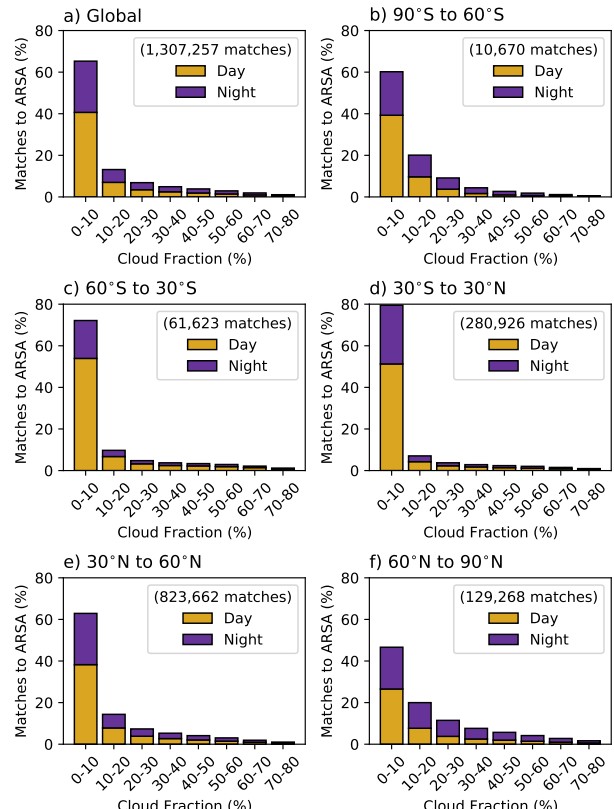

**Figure 12.** As Figure 9, binned sampling of IMS collocations over ARSA sites for global and split into the five latitudinal bands used in this study. The total number of collocations is shown in the legend for each subplot.

spheric layer, the magnitudes are similar though reversed in sign. This suggests that cloud dependence observed in the global results has no diurnal dependence, whereas the Southern Hemisphere does exhibit a diurnal dependence for bias changes to cloud fraction. The final region to note is found above 500 hPa in the Tropics, which shows the inverted general behaviour in a consistent manner across all cloud fraction amounts. Differences in diurnal biases at these altitudes are within 0.25 K of one another. Finally, the strongest diurnal bias differences to 'all' matches are found nearer the surface at all latitudes except for collocations above 60°N. The largest differences are observed between 850-400 hPa for cloud fractions below 40%. Relative to water vapour, temperature profile biases exhibit greater complexity in the presence of varying cloud amounts. As with GRUAN, collocations made over ARSA sites show disproportionate sampling under different cloud fraction amounts, with the highest frequency always seen for cloud fractions below 10% (Figure 12). Furthermore, daytime match-ups at ARSA sites dominate across all bins, whereas for GRUAN, it was nighttime collocations. With the separation of the ARSA results into the five latitudinal bands, any sampling bias will have a

**Table 3.** Stability of IMS water vapour profile biases at ARSA sites, given as the trend in the bias, with units of % ppmv/decade. Trends are reported for all global matches, split between all matches, daytime and night-time only IMS soundings, and a breakdown for each of the five broad latitude bands. It should be noted that the separation between day and night cases is more representative of seasonal differences (e.g. polar winter/summer) than diurnal ones for high latitudes. Values denoted with * are outside the 95% confidence interval.

| Pressure layer | Global | 90°S-60°S | 60°S-30°S | 30°S-30°N | 30°N-60°N | 60°N-90°N |
|---|---|---|---|---|---|---|
| **All soundings** | | | | | | |
| 400-300 hPa | 0.79±0.83 | -10.71±4.45* | -0.29±0.73 | 0.57±0.66 | 2.04±0.99 | -0.29±1.10 |
| 500-400 hPa | -1.76±1.19 | -10.18±5.36* | -2.65±1.64 | -2.52±1.23 | -0.38±1.32 | 0.56±1.41 |
| 700-500 hPa | -1.34±1.34 | 0.11±4.91* | -3.32±1.55 | -2.23±1.03 | -0.74±1.38 | 1.20±1.81 |
| 850-700 hPa | 0.93±1.01 | -5.94±4.63* | 0.74±1.40 | -1.32±0.89 | 2.16±1.62 | 3.38±1.84 |
| 925-850 hPa | 0.21±0.69 | 0.97±3.34* | 1.42±1.09 | -1.22±0.78 | 0.86±0.97 | 1.41±1.22 |
| 1000-925 hPa | -0.25±0.91 | 2.61±3.53* | 1.31±0.93 | -2.31±1.05 | 0.61±0.84 | 0.96±1.26 |
| | | | | | | |
| **Daytime soundings** | | | | | | |
| 400 to 300 hPa | 1.16±0.95 | -7.38±6.69* | -0.38±0.88 | 1.27±0.63 | 1.84±0.85 | 2.00±2.02 |
| 500 to 400 hPa | -2.46±1.35 | -4.76±8.29* | -2.49±1.64 | -0.66±0.54 | -2.68±1.55 | 0.95±2.40 |
| 700 to 500 hPa | -2.45±1.66 | -1.72±9.17* | -3.88±1.67 | -1.47±0.74 | -2.67±1.40 | 2.04±2.45 |
| 850 to 700 hPa | 1.30±1.38 | -11.37±7.24* | 0.45±1.47 | 0.06±0.46 | 1.73±1.71 | 3.11±1.80 |
| 925 to 850 hPa | 1.09±1.12 | 2.14±4.27* | 1.49±0.95 | 0.80±1.07 | 1.13±1.09 | 1.35±1.78 |
| 1000 to 925 hPa | 0.81±0.94 | 3.01±4.48* | 1.89±0.95 | -0.41±0.73 | 1.50±1.15 | -0.75±2.04 |
| | | | | | | |
| **Night-time soundings** | | | | | | |
| 400 to 300 hPa | 0.38±1.17 | -7.54±6.89* | 0.51±1.15 | -0.74±1.10* | 2.36±1.25 | -0.35±1.14 |
| 500 to 400 hPa | -0.55±1.14 | -17.96±8.26* | -1.74±1.83 | -5.81±2.30 | 2.94±1.18 | 1.85±1.56 |
| 700 to 500 hPa | 0.42±0.83 | -3.17±6.85* | -2.29±2.01 | -3.85±1.85 | 2.25±1.66 | 2.25±1.84 |
| 850 to 700 hPa | 0.45±0.59 | -6.24±7.27* | 2.71±1.64 | -3.45±1.65 | 2.76±1.31 | 3.10±1.65 |
| 925 to 850 hPa | -0.83±0.85 | 11.22±12.29* | 1.44±1.28 | -4.89±2.14 | 0.73±1.31 | 2.06±1.33 |
| 1000 to 925 hPa | -1.57±1.10 | 8.11±9.02* | 0.21±0.90 | -5.98±3.08 | -0.34±1.11 | 0.76±1.28 |

higher impact due to the lower collocation numbers outside of 30°N-60°N.

## 5.3 Bias Stability

The final analysis performed on IMS profiles looks at the stability of the biases over the study period (June 2007- December 2017). The Global Climate Observing System (GCOS) sets performance requirements for essential climate variables (ECVs), which for water vapour and temperature profiles are 0.3 %/decade and 0.05 K/decade respectively (GCOS, 2016). It should be noted that the unit for water vapour is in absolute units (e.g. ppmv) rather than relative humidity (% RH). Monthly median biases were first calculated for global and latitudinal banded results, and the trends were calculated using equation 18. For the linear trend model, we follow the same approach used within the Global Energy and Water Exchanges (GEWEX) water vapor assessment (G-VAP) is followed where four frequencies and the strength of the El Niño Southern Oscillation are fitted simultaneously as part of the regression. A correction is also applied to the trend uncertainty for autocorrelation (for details, please see Schröder et al. (2019)). While the record analysed is only 9.5 years, results are scaled to give stability in units per decade for comparison to GCOS requirements.

Results for water vapour profile bias stability are given in Table 3. Global comparisons show that biases between 850 hPa and the surface are within 0.3 % ppmv/decade for all cases (global average). However, when the split between day and night matches is examined, we observe positive trends for daytime cases and negative trends for night-time cases, all outside GCOS requirements. Above this altitude, the bias trend increases and switches sign, before reducing in the upper tropospheric layer and becoming positive again. This behaviour is driven more by the daytime collocations, whilst the night-time cases generally show smaller positive bias trends (i.e. stability) relative to all matchups.

When broken down into the five latitude bands, stability performance becomes more complex, with broad variability for the calculated trends ranging from -10.71±4.45 to 3.38±1.84 % ppmv/decade. When split diurnally, this range increases to between -17.96±8.26 and 11.22±12.29, with both maximum and minimum found between 90°S-60°S from night-time cases. Negative trends between 850-1000 hPa observed in the tropics influence global results and are driven by daytime cases. A key point to note is that though several signals are fitted to the bias time series, a significant amount of noise remains. Only 53% of results are outside the uncertainty, and these tend to be where lower performance is observed, e.g. 90°S-60°S. Night-time trends have a slightly higher performance, with 60% of the trends outside of the uncertainty. While exhibiting strong negative and positive gradients, the bias trends for the Antarctic latitudinal band have little impact on the global result. This is reassuring as they are all outside the 95% confidence interval. For results within the 95% confidence interval, the uncertainties on water vapour bias trends range from 0.66-1.84 % ppmv/decade for all cases, 0.46-2.45 % ppmv/decade for daytime only, and 0.9-2.3 % ppmv/decade for night-time match-ups.

Table 4 gives the bias trends for global and latitudinal band match-ups for IMS temperature profiles. Trends range between -0.41±0.36 and 0.31±0.24 K, with only two layers in the mid-to-upper troposphere between 90°S-60°S falling within GCOS requirements. Examination of the day and night-time trends show that these polar values are dominated by daytime cases. The two surface layers in the global results are close to the 0.05 K/decade requirement for all matchups, which is driven by night-time surface trends between ±60°. However, where we find small stability trends (higher performing), they are inside of the trend uncertainty. Similar to the water vapour results, we find that only 47% of trends are outside the noise for all daytime cases, with a slight improvement in night-time matchups where 50% of the trend are outside the uncertainty. The vertical pattern in global trends is matched in latitudinal results between 60°S to 60°N, with positive trends between 1000-850 hPa and negative trends above 850 hPa. At polar latitudes, tropospheric temperature bias trends are predominately positive, with the stronger gradients observed in the Northern Hemisphere. The uncertainty range for all trends is between 0.06 and 0.98 K/decade, with polar night-time match-ups dominating the higher end of this range. While the temperature profile stability is not quite meeting GCOS requirements, it is worth noting that daytime polar (SH) and night-time northern mid-latitudes both have 4 out of 6 layers with bias trends of 0.06 K/decade or less.

**Table 4.** As Table 3 but for temperature profile biases. Trends are given in K/decade.

| Pressure layer | Global | 90°S-60°S | 60°S-30°S | 30°S-30°N | 30°N-60°N | 60°N-90°N |
|---|---|---|---|---|---|---|
| **All soundings** | | | | | | |
| 400-300 hPa | -0.12±0.12 | -0.18±0.15 | -0.18±0.12 | -0.26±0.16 | -0.11±0.09 | 0.31±0.24 |
| 500-400 hPa | -0.23±0.21 | 0.00±0.38 | -0.28±0.11 | -0.33±0.31 | -0.17±0.25 | 0.28±0.79 |
| 700-500 hPa | -0.32±0.18 | 0.03±0.22 | -0.34±0.15 | -0.41±0.36 | -0.28±0.13 | 0.19±0.87 |
| 850-700 hPa | -0.14±0.21 | 0.22±0.39 | -0.13±0.16 | -0.21±0.14 | -0.12±0.27 | 0.27±0.97 |
| 925-850 hPa | 0.08±0.15 | 0.26±0.49 | 0.13±0.18 | 0.11±0.24 | 0.07±0.17 | 0.30±0.59 |
| 1000-925 hPa | 0.10±0.27 | 0.15±0.41 | 0.28±0.18 | 0.12±0.10 | 0.13±0.24 | 0.31±0.28 |
| | | | | | | |
| **Daytime soundings** | | | | | | |
| 400 to 300 hPa | -0.14±0.13 | -0.11±0.23 | -0.23±0.14 | -0.24±0.14 | -0.14±0.11 | 0.24±0.18 |
| 500 to 400 hPa | -0.24±0.13 | 0.06±0.39 | -0.28±0.10 | -0.28±0.23 | -0.20±0.21 | 0.26±0.53 |
| 700 to 500 hPa | -0.36±0.18 | 0.01±0.33 | -0.38±0.16 | -0.38±0.29 | -0.37±0.13 | 0.22±0.44 |
| 850 to 700 hPa | -0.17±0.17 | -0.01±0.53 | -0.18±0.22 | -0.22±0.14 | -0.20±0.28 | 0.37±0.53 |
| 925 to 850 hPa | 0.13±0.22 | -0.01±0.60 | 0.11±0.27 | 0.18±0.31 | 0.08±0.32 | 0.43±0.38 |
| 1000 to 925 hPa | 0.17±0.28 | -0.11±0.48 | 0.22±0.23 | 0.25±0.09 | 0.16±0.25 | 0.33±0.21 |
| | | | | | | |
| **Night-time soundings** | | | | | | |
| 400 to 300 hPa | -0.09±0.08 | 0.46±0.35 | -0.09±0.07 | -0.30±0.13 | -0.06±0.06 | 0.36±0.21 |
| 500 to 400 hPa | -0.20±0.29 | 0.44±0.38 | -0.26±0.10 | -0.42±0.31 | -0.12±0.27 | 0.35±0.79 |
| 700 to 500 hPa | -0.24±0.29 | 0.44±0.30 | -0.26±0.08 | -0.46±0.32 | -0.17±0.24 | 0.27±0.96 |
| 850 to 700 hPa | -0.07±0.41 | 0.62±0.47 | -0.14±0.09 | -0.17±0.11 | -0.03±0.43 | 0.30±0.98 |
| 925 to 850 hPa | 0.03±0.13 | 0.59±0.73 | 0.02±0.08 | -0.03±0.24 | 0.06±0.14 | 0.31±0.54 |
| 1000 to 925 hPa | 0.00±0.23 | 0.46±0.70 | 0.21±0.28 | -0.15±0.68 | 0.06±0.40 | 0.29±0.27 |

## 6 Discussion

The EUMETSAT Polar System (EPS) programme, which began in 2006 and will run until 2027, consists of the Meteorological operational satellite (Metop) series of platforms; Metop-A, Metop-B and Metop-C. Continued through the EPS Second Generation programme, the data record from this satellite series will provide a continuous data record out to 2045. Therefore, Metop data products are invaluable for climate data records (CDRs). IASI water vapour and temperature profile data analysis has been performed from four additional sources. The main operational source comes from the European Organisation for the Exploitation of Meteorological Satellites (EUMETSAT), with the other datasets available from National Oceanic and Atmospheric Administration (NOAA), Laboratoire atmosphéres, milieux, observations spatiales (LATMOS)/Université libre de Bruxelles (ULB; https://iasi-ft.eu/products/atmospheric-temperature-profiles/), and Karlsruhe Institute of Technology (KIT) as part of the MUlti-platform remote Sensing of Isotopologues for investigating the Cycle of Atmospheric water (MUSICA) project (Schneider et al., 2016).

With our study, we have adopted a methodology to that outlined in Trent et al. (2019) so that the results are comparable within a common framework. This is especially important for future validation exercises when considering combining different platforms to create a CDR. Capability from other international collaborative efforts has been incorporated to extend the scope of this validation framework. Tools developed within G-VAP (Schröder et al., 2019) have been employed to investigate the stability of observed biases, which is vital for the long-term characterisation of CDR performance.

Early assessment of the EUMETSAT operational processor L2 during the Joint Airborne IASI Validation Experiment (JAIVEx) revealed Relative Humidity (RH) profiles to be dry biased near the surface (-10% RH) and at altitudes above 12 km (-10 to -5% RH), with wet biases in the free troposphere reaching 5% RH. Temperature profiles were shown to be cold-biased and within 1 K of radiosonde measurements below 12 km (Zhou et al., 2009). A second presented in Pougatchev et al. (2009) looked at 650 collocations over Lindenberg, Germany. IASI RH profiles showed a predominate wet bias within 10% RH between 800-300 hPa, with a maximum of 20% RH observed between 300-200 hPa. Temperature profile biases were shown to exceed 2 K near the surface. In contrast, between 950-100 hPa, biases oscillate between ±0.5 K. Further analysis of EUMETSAT IASI water vapour over the Tibetan Plateau revealed profile biases within 25% g/kg of RAOB radiosondes, with RMS differences between 20-50% g/kg (Ting et al., 2013). Differences in observed bias strongly depended on seasonality, with warmer months generally wet-biased at altitudes lower than 600 hPa, while colder months exhibited a dry bias. While August et al. (2012) present results of temperature and water vapour profile comparisons to reanalysis, they are not included as part of this discussion as we consider these to be inter-comparisons rather than validation. The most recent validation results for the operational IASI product can be found in the L2 validation report (EUMETSAT, 2018). Here, comparisons to global radiosondes show temperature profile biases within ±0.5 K (predominately cold biased) and water vapour profiles are within 0.1-0.2 g/kg, dry biased in the lower troposphere and wet biased at altitudes above 800 hPa.

Divakarla et al. (2011) compared NOAA-processed IASI/MHS and AMSU-A water vapour and temperature profiles to RAOB radiosonde soundings using the same collocation criteria as our study. Match-ups were split into four groups; i) clear-sky ocean, ii) clear-sky all surfaces, iii) cloud-cleared ocean, and iv) cloud-cleared all surfaces. The term cloud cleared refers to the method outlined in Susskind et al. (2003) for treating IR measurements in cloudy scenes when used in conjunction with MW radiances. Results showed RMS differences of between ≈15-40% RH and 1-1.5 K between the surface and 300 hPa for water vapour and temperature, respectively. Temperature profiles were shown to have slightly higher performance in clear-sky scenes, while water vapour had better accuracy in cloud-cleared cases.

Further analysis of NOAA IASI product over East Asia Kwon et al. (2012) showed relative humidity biases with respect to RAOB radiosondes are within ±5% RH over all surfaces except land, where biases rise to ≈8% RH. Temperature profile biases were ±0.5 K between the surface 800-200 hPa. Nearer the surface, temperature profile biases can exceed 2 K. Wet, and warmer biases were observed in drier atmospheres, and similar performance was seen across differing cloud fraction amounts.

The study from Bouillon et al. (2022) presents the most comprehensive analysis of IASI temperature pro-

files by comparing 13 years of retrievals using an artificial neural network (ANN) with ARSA, European Centre for Medium-Range Weather Forecasts (ECMWF) reanalysis (ERA5; (Hersbach et al., 2020)), and the climate data record (CDR) of allsky IASI temperature profiles from EUMETSAT (released 2020; https://doi.org/10.15770/EUM_SEC_CLM_0027). Results from this study found; i) good agreement between all four data sets, especially between 7-750 hPa, ii) differences between ARSA and ERA5 were less than 0.5 K over the majority of latitudes and pressure levels, and iii) a warming trend in tropospheric temperature of 0.5 K/decade at mid-latitudes and 1 K/decade at the poles due to Arctic amplification. Finally, Validation of MUSICA IASI (IR only) water vapour profiles at selected GRUAN sites showed biases with 11% below 12 km; a stronger dry bias was observed at higher altitudes of up to 21% ppmv Borger et al. (2018). With the MUSICA IASI scheme only using IR information, these biases represent clear skies only.

Looking forward, it is important that common units and metrics are adopted for tropospheric profile analysis. This is especially true for water vapour profiles due that the range of units that can be used, both absolute and relative. For atmospheric temperature, we see similar results to EUMETSAT (2018) and Bouillon et al. (2022) with profiles predominantly cold-biased and below 0.5 K. For water vapour we can only really compare to results from Borger et al. (2018), where we see a similar performance below 12 km (within 11% ppmv). With our study, we have gone further to look at the impacts of clouds and diurnal sampling on the report biases. Furthermore, we present the first results on the stability of these biases, values that are needed when considering using the data for climate applications.

## 7   Conclusions

This study has assessed 9.5 years of IMS water vapour and temperature profiles co-retrieved from IASI, MHS and AMSU-A onboard the Metop-A platform. This dataset was produced as part of the ESA Water Vapour Climate Change Initiative (WV_cci). A database of match-ups was collected over GRUAN and ARSA radiosonde sites for IASI footprints within 100 km and $\pm 3$ hrs of launch. These broad collocation criteria allow multiple IMS profiles to be averaged over each site to reduce the collocation uncertainty. IMS averaging kernels were applied to all matched radiosonde profiles, smoothing the higher (vertical) resolution *in situ* measurements to the satellite vertical atmospheric sensitivity, thus allowing for like-for-like comparisons. Evaluation of IMS performance was conducted for 'global' matches, day/night differences, the impact of cloud fraction, and site-to-site differences. From the results gathered, the main conclusions from this study are:

- Global biases calculated from 9.5 years of matches made at both GRUAN and ARSA sites are within performance goals of 10 % ppmv and 1 K for water vapour and temperature profiles respectively (Hilton et al., 2012). The strongest wet biases for collocations made at GRUAN sites can be found in the upper troposphere, while the lower tropospheric layers exhibit the wetter biases for ARSA stations. Temperature profiles are predominately cold-biased relative to ARSA. In contrast, a small warm bias is found for GRUAN comparisons in the mid-to-upper troposphere.

- Although global results are within desired performance limits, site-to-site differences are observed for GRUAN sites. Likewise, biases reported as a function of latitude relative to ARSA profiles increases in magnitude, moving towards higher latitudes. Biases are larger at latitudes $>60°$ than at 30-60° though within respective 1% and 1 K limits, except for water vapour at latitudes $>60°$N. Some of this variability can be explained by sampling differences, especially for GRUAN, where a difference of three orders of magnitude between numbers of match-ups can be found. However, the ARSA results indicate a latitudinal dependence of biases related to total column concentrations (e.g. Roman et al. (2016)).

- Water vapour profiles bias for global GRUAN matches between the surface and 850 hPa re substantially larger in the daytime, with corresponding night-time biases within $\pm 0.3$ % ppmv. However, night-time biases exceed daytime biases in layers above 700 hPa. Temperature biases in layers up to 500 hPa are warm-biased at night and cold-biased for daytime matches. Overall, night-time temperature profile biases dominate the global results.

- Globally, biases from ARSA match-ups have consistent diurnal patterns with those made at GRUAN sites. However, daytime data dominates (cold) temperature biases rather than night-time match-ups. On latitude bands, this daytime dominance in temperature profile biases is driven by tropical and mid-latitude collocations.

- Further to the diurnal (polar summer/winter) effects on reported biases, cloud fraction as has an additional impact. For water vapour profile match-ups, daytime water vapour biases relative to ARSA tend to be wetter than global (daytime) results under all cloud fraction amounts. Similarly, cloudy night-time matches are drier the those shown in Figure 11d. This pattern is consistent for matches between $\pm 60°$, although it breaks down for polar sites. For GRUAN collocations, a slightly different pattern is observed. Profiles become drier relative to global results with increasing cloud fraction in the mid-to-lower troposphere, with the upper troposphere biases consistently wetter. This pattern is inverted for night-time match-ups. Cloud fraction impacts on temperature

biases have better global consistency with cloudy scenes colder for daytime and warmer for night-time for both GRUAN and ARSA match-ups. However, as is shown by Figures 9 & 12 these results will include a sampling bias due to the relatively low numbers of matches over the differing cloud fraction amounts relative to cloud fraction below 10%. This feature in the data can partially be attributed to the fact that not every retrieval in the IMS product contains an averaging kernel, so we are limited in selecting profiles we can use in this study before quality filtering.

– Initial look into the height-resolved stability of the IMS biases (with respect to ARSA) over the 9.5-year time series shows that most are outside of GCOS requirements. However, to the authors' knowledge, this is the first look at profile bias stability from combined microwave and IR nadir sounders. Therefore, it is understood that while a longer time series is required to conduct this analysis, a greater understanding of the impact of collocation uncertainties on the trend is also needed. It is anticipated that this can explain the effects seen in mid-tropospheric values (Tables 3 & 4).

Finally, results from this study show the promise of satellite water vapour and temperature profile records for long-term climate studies, especially for scenes with cloud fractions below 0.1. Additionally, knowledge of temperature and water vapour profile accuracy is vital for calculating uncertainty budgets for all trace gases retrieved from IASI (like) data. Therefore, the extension of combined IR+MW profile records forward in time through MetOp-B/C, MetOp-SG and the US series S-NPP/JPSS (NOAA-20 onwards), but also backwards with ATOVS, is needed. A data set of this type would take the time series back to 1999 and out to 2045. Adding new missions into the time series will also capture greater diurnal variability for profiles and TCWV. This final point would provide a new complimentary record for the existing SSM/I (SSMIS) record (ice-free oceans only), which could be used to extend coverage over land and possibly polar regions.

*Data availability.* The IMS Version-1 dataset is now archived at the Centre for Environmental Data Analysis (CEDA) http://dx.doi.org/10.5285/489e9b2a0abd43a491d5afdd0d97c1a4 and made available to the European Space Agency Water Vapour Climate Change Initiative project http://cci.esa.int/watervapour. GRUAN data is freely available from the Copernicus Climate Data Store (CDS) https://cds.climate.copernicus.eu/cdsapp#!/dataset/insitu-observations-gruan-reference-network?tab=overview, while the ARSA radiosonde archive is accessible via registration at https://ara.lmd.polytechnique.fr/index.php?page=arsa-registration.

## Appendix A: GRUAN Site details

Information on the GRUAN radiosonde sites available at the time of this study are detailed in Table A1. Coincident data was found for all sites except Darwin (DAR) for the study time period June 2009 to December 2017.

**Table A1.** Details of GRUAN sites used by this study. Station information has been taken from https://www.gruan.org/network/sites.

| Code | Name | Latitude | Longitude | Altitude |
|------|------|----------|-----------|----------|
| BAR | Barrow, AK, USA | 71.32° | -156.61° | 8 m |
| BEL | Beltsville, MD, USA | 39.05° | -76.88° | 53 m |
| BOU | Boulder, CO, USA | 39.95° | -105.20° | 1743 m |
| CAB | Cabauw, Netherlands | 51.97° | 4.92° | 1 m |
| DAR | Darwin, Australia | -12.43° | 130.89° | 30 m |
| GRA | Graciosa, Portugal | 39.09° | -28.03° | 30 m |
| LAU | Lauder, New Zealand | -45.05° | 169.68° | 370 m |
| LIN | Lindenberg, Germany | 52.21° | 14.12° | 98 m |
| MAN | Manus, Papua New Guinea | -2.06° | 147.42° | 6 m |
| NAU | Nauru, Nauru | -0.52° | 166.92° | 7 m |
| NYA | Ny-Ålesund, Norway | 78.92° | 11.93° | 5 m |
| PAY | Payerne, Switzerland | 46.81° | 6.95° | 491 m |
| POT | Potenza, Italy | 40.60° | 15.72° | 720 m |
| REU | La Réunion, France | -20.89° | 55.49° | 13 to 2200 m |
| SGP | Lamont, OK, USA | 36.60° | -97.49° | 320 m |
| SOD | Sodankylä, Finland | 67.37° | 26.63° | 179 m |
| TAT | Tateno, Japan | 36.06° | 140.13° | 27 m |
| TEN | Tenerife, Spain | 28.32° | -16.38° | 115 m |

## Appendix B: Sampling over GRUAN sites in low cloud fractions

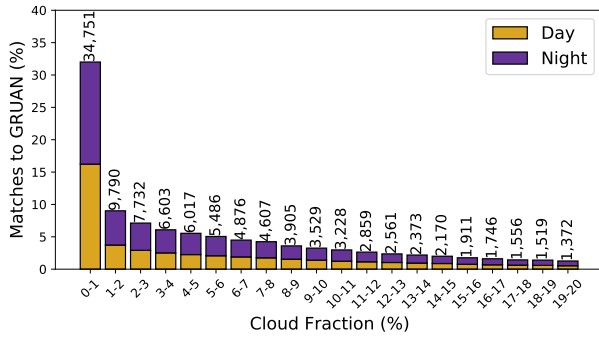

**Figure B1.** As Figure 9, stacked histogram of IMS collocations to GRUAN sites split into 1% cloud cover intervals between 0-10%. Again, results are separated into day and night cases with the total number of matches shown at the top of each bar.

To illustrate the disparity between the number of near clear-sky collocations and those under increasing cloud faction, the match-up statistics from Figure 9 for the first bin (0-10%) can be further broken down into 1% cloud fraction

bins. Shown in Figure B1, it can be seen that approximately 40% relative of match-ups in the 0-10% bin can be attributed to clear skies (0-1%).

*Author contributions.*  TT and JR designed the original work package within ESA WV_cci project, which partially funded this study. TT conducted the validation work and wrote the first draft of the manuscript. BC and RS produced and supplied the IMS data as well as provided expertise on its use. NS provided the ARSA profiles and input on their use and limitations. MS provided input and detail surrounding the ATOVS record. All authors contributed to discussions on the results and revisions of the manuscript.

*Competing interests.*  The authors declare that there are no competing interests.

*Acknowledgements.*  This study was partly funded by ESA (Contract No. 4000123554) via the Water_Vapour_cci (WV_cci) project of ESA's Climate Change Initiative (CCI). Tim Trent, Brian Kerridge, Richard Siddens and John Remedios would also like to acknowledge the funding from Natural Environment Research Council through Natural Centre for Earth Observation, contract PR140015. This research used the ALICE High-Performance Computing Facility at the University of Leicester. Marc Schröder acknowledges the financial support of the EUMETSAT member states through CM SAF. The authors would like to thank the ARA/LMD group for the production, validation and availability of the ARSA (Analyzed RadioSounding Archive) database. The ARA / LMD group gratefully acknowledges the ECMWF data server for making available the ERA-interim outputs, the radiosonde archive, and the surface station archive that comprises the ARSA database. The authors would also like to thank the two reviewers who provided very useful and constructive comments that helped refine the final manuscript.

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
