# Peer review of "Evaluation of tropospheric water vapour and temperature profiles retrieved from Metop-A by the Infrared and Microwave Sounding scheme"

_EGUsphere, 2022_

## Author Comment (AC1)

**Authors' Response to Reviews of**

**Evaluation of tropospheric water vapour and temperature profiles retrieved from Metop-A by the Infrared and Microwave Sounding scheme**

Tim Trent, Richard Siddens, Brian Kerridge, Marc Schrörder, Noëlle A. Scott, and John Remedios
*EGUsphere,* `https://doi.org/10.5194/egusphere-2022-757, 2022`
* * *
**RC:** *Reviewers' Comment*,     AR: Authors' Response,     ☐ Manuscript Text

**1.  Reviewer #1**

**RC:**  *The Meteorological Operational satellite (Metop) series of platforms operated by the European Organisation for the Exploitation of Meteorological Satellites (EUMETSAT) have provided valuable observations of the Earth's surface and atmosphere for meteorological and climate applications. These datasets will provide a continuous data record out to 2045. Therefore, Metop data products are an invaluable source for climate data records (CDRs). The authors present a comprehensive assessment of profile data produced using the Infrared and Microwave Sounding (IMS) scheme with the European Space Agency (ESA) Water Vapour Climate Change Initiative (WV_cci) against radiosondes from the Global Climate Observing System (GCOS) Reference Upper-Air Network (GRUAN) and Analysed Radio Soundings Archive (ARSA) data records, and found that the results from this study demonstrate the real potential for tropospheric water vapour and temperature profile CDRs from the Metop series of platform. The manuscript is generally well-written and the scope is well-within the journal. I have two minor comments below, some focused on data visualization that I hope will help the authors as they consider a revision of their manuscript before recommending acceptance.*

**AR:**  We would like to thank the referee for taking the time to review our manuscript. Below we reply to the issues raised by the referee. The original reviewer comments (RC) are given in bold italics, with the author's responses (AR) in plain text. Where we have updated the manuscript, the extract is included in a quote box with the original removed text in red and struck out. New text appears in blue and is underlined.

**1.1.**

**RC:**  *First, I don't learn more about the Metop series of platform, but I think it would be better to show global distributions of tropospheric water vapour and temperature profile CDRs from the Metop data against the ARSA or ERA5 reanalysis, which can help us see how well the Metop data match other references for a global scale.*

**AR:**  The scope of the paper is the analysis of only Metop-A, which is why we do not discuss Metop-B/C or the upcoming Metop-SG series of platforms.

Regarding the suggestion about showing global distributions of IMS water vapour and temperature profiles, we propose including a new figure that shows the differences between IMS and ERA5. The distribution of ARSA profiles does not lend itself to such a plot. Therefore, we show the differences for IMS profiles matched to ERA5 for both temperature and water vapour as a latitudinal average. In addition, this figure includes the standard deviation for the latitudinal means:

[Figure]

Figure 1: Example of the global mean differences between IMS temperature and water vapour profiles and ERA5 reanalysis for the 15$^{th}$ June 2012. Also included are the standard deviations (Stddev) for the differences. Reanalysis has been interpolated to the observation time and the centre of the IASI instantaneous field view. Before differences were calculated, the IMS averaging kernels were applied to the reanalysis profiles. For further discussion on averaging kernels, refer to Section 3 (Methodology).

To support this figure, we also propose updating the text:

> This study evaluates a 9.5-year record of temperature and humidity profiles from IASI and its companion MW instruments onboard Metop A, retrieved using the RAL IMS scheme ,and produced as part of the European Space Agency (ESA) Water Vapour Climate Change Initiative (WV_cci). Validation of this While modern NWP systems assimilate some spectral information from IASI and other satellites, the IMS product is designed to be independent of reanalysis. Therefore, in addition to climate model evaluation, tropospheric profile information from IMS can be used for comparative studies of reanalysis for both meteorological and climate applications. This is especially true for geographic regions which little or no *in situ* information with which to constrain the reanalysis. An example of this application is shown in Figure 1, where ERA5 has been collocated with IMS water vapour and temperature profiles. Here we see the daily differences between the satellite and reanalysis, with the biggest differences observed over polar regions. The assertion here is that the IMS will look to maximise information content from each set of measurements in a way that is too computationally expensive for reanalysis. However, for users to be confident of the use of IMS in such a manner, profiles need to be validated so that their performance is characterised.

**1.2.**

**RC:** *Second, the reference data, i.e., the GRUAN and ARSA also have certain biases. The differences between them would be better to be addressed somewhere in this manuscript.*

AR: We agree that the differences in bias and limitations of each record may not be clear to readers unfamiliar with the these radiosonde archives. Therefore, we propose updating section 2.2 to clarify these points:

[revised manuscript text omitted]

---

## Author Comment (AC2)

**Authors' Response to Reviews of**

**Evaluation of tropospheric water vapour and temperature profiles retrieved from Metop-A by the Infrared and Microwave Sounding scheme**

Tim Trent, Richard Siddens, Brian Kerridge, Marc Schrörder, Noëlle A. Scott, and John Remedios
*EGUsphere,* `https://doi.org/10.5194/egusphere-2022-757, 2022`
* * *
**RC:** *Reviewers' Comment*,  AR: Authors' Response,  ☐ Manuscript Text

**1. Reviewer #2**

**RC:** *This paper describes the validation of the RAL IMS water vapor and temperature profiles retrieved from infrared hyperspectral and microwave sounders. This is useful if this dataset is to be used as a CDR for climate applications. The paper is well written and well structured. I therefore deserves publishing.*

AR: We would like to thank the referee for taking the time to review our manuscript. Below we reply to the issues raised by the referee. The original reviewer comments (RC) are given in bold italics, with the author's responses (AR) in plain text. Where we have updated the manuscript, the extract is included in a quote box with the original removed text in red and struck out. New text appears in blue and is underlined.

**1.1. Page 3 line 63. Section 2.Data**

**RC:** *The retrievals are not only IASI based, so you might want to add something like "and microwave sounder data".*

AR: We thank the reviewer for highlighting this point. Indeed, the IMS scheme uses infrared and microwave sounder data to retrieve water vapour and temperature profiles. Therefore, we have updated the text to make sure this point is clear:

> This section describes the algorithm used to retrieve water vapour and temperature profiles from IASI  with companion microwave sounder data and details of the radiosonde data sets used for their assessment.

**1.2. Page 4. line 99.**

**RC:** *You might want to add something like "This is justified by the finding that WV inhomgeneities within the FOV do cause a significant modification in the results of the radiative transfer modeling (`https://amt.copernicus.org/articles/11/6409/2018/`,`https://cimss.ssec.wisc.edu/itwg/itsc/itsc23/presentations/oral.2.01.calbet.pdf`).*

AR: We have added the reference and a sentence to reflect the links between the two bodies of work. Inhomogeneities and their impacts will have an increasing significance as a source of uncertainty for profile climate data records. Therefore it is correct to acknowledge the previous study here.

> The bias correction is needed to account for systematic differences between RTTOV and the IASI observations, including errors in RTTOV. Allowing the retrieval to fit scale factors $x_{b0}$ and $x_{b1}$, instead of assuming a fixed scan-angle dependence improves the fit (gives lower cost) over a wide range of observing conditions. The recent study by Calbet et al. (2018) supports this approach as they demonstrated that the inhomogeneities in water vapour within a satellite Instantaneous Field of View (IFOV) cause a significant modification in the results from radiative transfer modelling.

**1.3.**

**RC:** *I would be interested in seeing the results of the bias corrections. If the inhomogeneity correction hypothesis is true, the biggest contribution to this bias will come from inhomegeneities in WV within the FOV. So this could be a way to measure them. They potentially can be correlated with turbulence.*

**AR:** We thank the reviewer for raising this interesting point. Firstly, scene inhomogeneity across IASI IFOVs is accounted for by EUMETSAT in generating L1 data through the use of co-located AVHRR images. This inhomogeneity is predominantly due to the presence of clouds. For the retrieval, water vapour spectral features present in the systematic bias spectra are most likely due to inconsistencies in H2O spectroscopic data between different wavelength intervals used by IMS. Though not impossible that scene inhomogeneity might contribute in certain cases, this would require a detailed study which would not be merited at this stage in our view. To reflect the points we have added additional text to the manuscript and direct the reader to examples of the bias-corrected spectra from RTTOV that can be found in the EUMETSAT report, Siddans & Gerber, 2015.

> The bias correction is needed to account for systematic differences between RTTOV and the IASI observations, including errors in RTTOV. Allowing the retrieval to fit scale factors $x_{b0}$ and $x_{b1}$, instead of assuming a fixed scan-angle dependence improves the fit (gives lower cost) over a wide range of observing conditions. Examples of these corrections to systematically biased spectra are given in Siddans & Gerber (2015). The recent study by Calbet et al. (2018) supports this approach as they demonstrated that the inhomogeneities in water vapour within a satellite Instantaneous Field of View (IFOV) cause a significant modification in the results from radiative transfer modelling. Observational inhomogeneities across the IASI IFOV are predominantly due to clouds within the scene. These effects are accounted for at the L1 data stage by EUMETSAT through the collocation of AVHRR images within the IFOV (EUMETSAT, 2019)

**1.4.**

**RC:** *I do not completely understand what DOFS is. And along with it Fig. 3. Also in Fig. 3 you say you plot DOFS but in the scale you show "% of profiles per bin". Please explain this better for people not familiar with DOFS. One or two sentences should be enough.*

**AR:** Firstly, we appreciate that the concepts around averaging kernels and degrees of freedom for signal (DOFS) may not be familiar to many people as they are not commonly used by the tropospheric water vapour community. Secondly, we thank the reviewer for highlighting possible confusion around Figure 3. Therefore, we propose to update the text (line 165 onwards) on averaging kernels and DOFS to make this point clearer:

> The practical uses of the square averaging kernel matrix (**A**)  are to i) smooth atmospheric profiles from models,

reanalysis and *in situ* measurements (discussed further in Section 3.2), and ii) to obtain the degrees of freedom for signal (DOFS) for specific retrieval products.  The DOFS are given by the trace (sum of the diagonal elements) of the sub-matrix of $\mathbf{A}$ corresponding to a specific product and represent the total number of independent pieces of information in the profile. The averaging kernel of retrieved water vapour profiles (defined on the RTTOV levels) with respect to perturbations on the fine atmospheric grid is given by:

$$\mathbf{A}_{qf} = \mathbf{M}_{qf}\mathbf{A}_{f:q}, \tag{10}$$

where $\mathbf{A}_{f:q}$ is the averaging kernel for the water vapour state vector elements with respect to perturbations in ln(ppmv) on the fine atmospheric levels. The averaging kernel for temperature is derived similarly using the corresponding matrices. An understanding of  a profile vertical resolution  can be inferred from the DOFS value, as it describes the number of independent pieces of information resolved (Rodger, 2000). Figure 3 shows the range of DOFS for IMS temperature and water vapour profiles as a function of latitude, with the tropopause height (TPH) overlaid.  The 2D histograms show the distribution of profile DOFS, from which we can see that in the tropics, most profiles sit between 6-7 and 11-12 DOFS for water vapour and temperature, respectively. Moving outwards through the mid-tropics to the high latitudes, the DOFS values reduce, with the distribution becoming more variable. Comparing the water vapour distribution to the cold point tropopause height (black dashed line), we can observe that they hold similar shapes, while for temperature, this is less so. This result is expected as nadir IR+MW sounders are predominately sensitive to the emissions from the troposphere.  , especially for water vapour.

The next conceptual step is how the DOFS relate to the vertical resolution of IMS profiles. Examples of averaging kernels for water vapour and temperature  profiles from the IMS L2 product are given in Figure 4.  , where we see that most of the information for water vapour is situated in the lowest 10 km of the atmosphere while for temperature is more continuous. Therefore, an examination of the cumulative degrees of freedom for signal (CDOFS) from these averaging kernels can be used to describe the vertical resolution of the retrieved profiles. The gradients of the CDOFS as a function of altitude can then be interpreted as the profile resolution at given heights. The desired performance for vertical resolutions from IASI is 1 & 2 km for temperature and water vapour profiles, respectively (Hilton et al., 2012). What can be seen from Figure 4 is that vertical resolution is not necessarily constant throughout the troposphere. Indeed, examining a different sounding over the same radiosonde site would show subtle differences. A key observation here is that the information from the water vapour profile terminates (vertical gradient) at the tropopause. Therefore, any use of the IMS water vapour profile above this height is meaningless.

To complement this new text, we also modify the caption of Figure 3 to orientate the reader better to the information given.

[Figure]

Figure 3: Visualisation of IMS water vapour and temperature profile degrees-of-freedom for signal (DOFS). This figure illustrates the latitudinal distribution of DOFS variability for both IMS water vapour and temperature profiles. DOFs were collected from the IMS L2 files between 2007-2016 and binned as a function of latitude. Values were then normalised using the total number of profiles in their respective latitude bin.  DOFs vary approximately between 2-7 for water vapour and 8-13 for temperature with strong peaks in the tropics. The spread in the data resembles the cold point tropopause height (TPH), especially for water vapour. The dashed black line represents the cold point TPH calculated from ERA5 temperature profiles (Hersbach et al., 2020).

**1.5.**

**RC:** *Please explain which radiosondes are used in GRUAN. There is a relatively big difference in WV between RS92 and RS41, being the latter much more precise.*

**AR:** In this study we only use the RS92 soundings. Therefore, we add a statement to the end of Line 189 to clarify this:

> An advantage of the higher resolution of GRUAN measurements is that it captures changes in humidity gradients and temperature inversions which can be missed or underrepresented by standard and significant pressure levels. It should be noted that the soundings from GRUAN feature only the Vaisala RS92 radiosondes measurements and not the more recent RS41.

**1.6. Page 10. line 133.**

**RC:** *I believe for $x_{(z)}$ you mean "layer mean profile" and not "weighted layers". This is confusing since in the next couple of sentences the term "layer mean profile" is used. Please correct or explain better.*

**AR:** The term $x_{(z)}$ in the text refers to the mean layer within a profile over which we are averaging. We thank the

reviewer for pointing out the confusion and acknowledge that the term needs to be updated to be consistent with others later in the text. Therefore, we propose to update the following text to clarify things for the reader:

> Next,  values are calculated for weighted layers ($\bar{\mathbf{x}}_{(z)}$) within each new layer mean profile ($\bar{\mathbf{x}}$), where the layer boundaries are defined by standard pressure levels defined at 1000, 925, 850, 700, 500, 400 and 300 hPa:
>
> $$\bar{\mathbf{x}}_{(z)} = \frac{\sum_{l=1}^{n} \mathbf{x}_{(l)} \mathbf{P}_{(l)}}{\sum_{l=1}^{n} \mathbf{P}_{(l)}}. \tag{13}$$
>
> Where $\mathbf{x}_{(l)}$ is the convolved radiosonde or IMS profile value at level $l$, $\mathbf{p}_{(l)}$ is the pressure profile value at level $l$, and $n$ is the numbers of levels in the layer.  Weighted layer mean profiles are not calculated for altitudes higher than 300 hPa because ARSA profile values are taken from ERA-Interim in the upper troposphere/stratosphere. All statistics used by this study are calculated from the layer mean profiles.

**1.7.**

**RC:** *Please note that RS92 sondes do not measure very well with low WV. This usually happens above tropopause. This is probably why the biases are much bigger at higher altitudes. You might want to repeat the statistics using only levels below tropopause. Or, equivalently, with small GRUAN uncertainties. This will most likely reduce the biases at high altitudes. Something to consider.*

**AR:** We thank the reviewer for this suggestion; this issue was something we were aware of when designing the analysis. The issue we faced is that the ARSA radiosondes do not have any uncertainty information associated with the profiles. Therefore, we could not apply the same criteria to these measurements. Experience from Trent et al. 2019 has shown us that for water vapour averaging kernels with sensitivity to the upper troposphere/lower stratosphere (UT/LS) actually have a bigger impact than the radiosonde biases due to the lack of sensitivity of IR sounder to moisture in this region. What we observe from Figure 2 is that above the tropopause the water vapour averaging kernels are very close to zero, and when applied to the difference between the radiosonde profile and a priori (equation 12) this term approaches also approaches zero. Therefore, above the tropopause the convolved profile essentially matches the a priori. This discussion is included with the updated text for averaging kernels (RC No. 1.4). Finally, the cut off of the profiles at 300 hPa also helps remove layers with UT/LS sensitivity or include profile levels above the tropopause.

**1.8. Page 10. line 206.**

**RC:** *Change "i) the dataset is being validated" with "i) the dataset being validated".*

**AR:** Done.

> The match-up database (MUDB) is generated by supplying the MUP a driver file containing information on i) the dataset  being validated,. . .

**1.9.**

**RC:** *Please explain in the text what MAD is when it first appears.*

**AR:** We have updated the text to clarify this point

> Therefore, we can think of the variability of the median as an estimate of the precision of the bias. To quantify the spread about the median, we calculate the median absolute deviation ($\sigma_{(z)}$), a robust measure of the data variability:
>
> $$\sigma_{(z)} = median|(\mathbf{x}_{(z)} - \mathbf{x}_{est(z)}) - b_{(z)}|. \tag{14}$$
>
> As we use robust statistics, the  median absolute deviation (MAD) values cannot be treated in the same way as standard deviation and used to calculate the standard error by dividing through by $\sqrt{N}$.

**1.10. Fig 5. and 6.**

**RC:** *If you use a smaller collocation window than 3 hrs and 100 km, the MAD will certainly disminish. Something to consider as en exercise.*

**AR:** Due to the static launch times of radiosondes when we reduce the number of sites with which we make collocations. This is especially true for the IMS product because we do not get averaging kernels for every IFOV. The scope of this study is to present the IMS product performance at global and sub-global scales, which requires collocation criteria that give enough scope to achieve this. Therefore, reducing these windows can introduce a significant sampling bias, with results becoming highly localised. Our strategy was to gather as many collocated pairs as possible and use robust statistics to minimise the impact of individual comparison performance arising from the broad collocation criteria. This approach works well in general, although suffers in areas of low coverage, as we would expect. We demonstrated this is a previous study and have added the reference to the text where we explain our strategy (lines 202-220):

> (line 216)...However, this is minimised by calculating global or per latitude band statistics which uses large numbers of matched pairs, unlike for site comparisons with a low number of matched cases. When using broad collocation criteria, any mismatch introduced during the match-up will affect the performance of individual comparison performance (Sun et al. (2010) & Sun et al.(2017)). Therefore, a robust statistics approach was adopted to minimise this effect as demonstrated in Trent et al. 2019.

**1.11. Fig. 8.**

**RC:** *Why is there less cases with higher cloud fractions? Can you give an explanation or hypothesis?*

**AR:** We thank the reviewer for raising this point as it was not clear in the manuscript. The combination of infrared and microwave instruments allows for the retrieval of profiles in the presence of cloud which would not be possible if the infrared alone were used. That said, cloud can still impact the retrieval and as such the product contains a flag that allows for cloudy scenes which the retrieval struggles to work for. Therefore, we see a disproportionate effect on scenes containing cloud especially this with higher cloud fractions. This results in the distribution we see in Figure 8. We propose adding this explanation to the text:

> Therefore, understanding the impact of cloud fraction within the IASI  IFOV on IMS profile biases is also of interest to this study. IMS profile biases were binned according to cloud fraction at intervals of 0.1 for all sites for all matches (day & night cases) and for the separate day and night cases. Water vapour and temperature bias results as a function of cloud fraction are presented in Figure 8 along with the difference between day and night cases to the 'all cases' result. It should be noted that a brightness temperature difference (BTD) flag is used to remove cloudy

scenes that significantly impact the retrieval. While IMS can produce profiles for cloudy IFOVs, the BTD flag will remove some of the profiles, disproportionately across increasing cloud fractions. This explains the distribution we observe in Figure 8.

**1.12.**

**RC:** *GRUAN WV measurements are bias corrected with, mainly, an estimation of the incident radiation from the Sun. This is critical for RS92 sondes. Not so much for RS41. Are ARSA sondes also bias corrected in WV?*

AR: ARSA is based on raw radiosondes observations extracted from the ECMWF archive. The required minimal information is to have radiosonde measured values from the surface up to 30 hPa for temperature profiles and from surface to 300 hPa for water vapor profiles. These radiosonde observations are extended above their highest measured point with time and space ERA-Interim co-located data up to 0.1 hPa(temperature, water vapour, and ozone up to 0.1 hPa) and then with SciSat ACE FTS level2 data (from 0.1hPa to the top the atmosphere: 0.0026 hPa). The validation of every surface to top ARSA profile relies upon the study of statistics (bias, standard deviation) between simulated and observed satellite radiances: TOVS and ATOVS, as well as, in more recent years, the MetOp A&B IASI, HIRS4, and MHS observations. The simulated data are generated by the 4A/OP radiative transfer model (Scott and Chédin, 1981, Chéruy et al, 1995) fed with the closest time and space ARSA colocated profile. Due to the excellent stability of the IASI radiances, the quality of the ARSA profiles may then be assessed. It is during such validation procedures that the water vapour profiles above 300hPa (i.e. in the area concerned by the extension of radiosondes by ERA-Interim co-located time-space profiles) turned out to be modified: the procedure is detailed in QUASAR (QUality Assessment of SAtellite and Radiosonde data) report (Scott, 2015). No correction is made to the water vapour due to an estimation of the incident radiation from the Sun".

**1.13.**

**RC:** *Are ARSA sondes RS92? Are they corrected from WV biases are they are in GRUAN (taking into account radiation)? Please explain in the paper. This would explain the day night bias difference in ARSA.*

AR: Because of its global coverage (see Figure 4) from nearly 1500 stations, ARSA includes RS92 but not only. No correction is made to the water vapour due to an estimation of the incident radiation from the Sun. The quality of ARSA profiles is assessed using match-ups with satellite radiance measurements with the development, validation, and characteristics of the ARSA database are fully described in the QUASAR (QUality Assessment of SAtellite and Radiosonde data) report (Scott, 2015). It is possible that this could explain the day night differences we observe between the two archives, though we do get different global sampling which will also impact the result. To clarify the differences between ARSA and GRUAN, we propose modifying the text on ARSA (line 190-199) to :

[revised manuscript text omitted]

**1.14.**

**RC:** *I would separate discussions and conclusions sections. The section is too long with too many comments for a conclusion.*

AR: We accept that the combined discussion and conclusion section benefits from being separated and have done so. To close off the new discussion section, we add the following text:

> . . . With our study, we have adopted a methodology to that outlined in Trent et al., (2019) so that the results are comparable within a common framework. This is especially important for future validation exercises when considering combining different platforms to create a CDR. Capability from other international collaborative efforts has been incorporated to extend the scope of this validation framework. Tools developed within G-VAP Schröder et al. (2019) have been employed to investigate the stability of observed biases, which is vital for the long-term characterisation of CDR performance.

However, we disagree that there are too many concluding points as this study performs an extensive analysis of temperature and water vapour profiles. Where previous studies will focus on one reference dataset, with a diurnal split and maybe some latitudinal analysis, we do this for two reference data records and investigate cloud effects and bias stability. Therefore, we feel the number of main conclusions (6) is appropriate. The final statement of this section was originally included as part of the main conclusions (bullet point); this has now been added to the main flow of the text, and the section flows better as a result.

**1.15.**

**RC:** *I would recommend to draw conclusions from night time sondes only. Since we know day time sondes have biases. Especially if they are RS92 sondes and even more if they are not bias corrected in ARSA. Your conclusions on bias trends might vary.*

AR: While we recognise that the nigh-time match-ups will give us the most robust results, we also feel that the 'all' and 'daytime' results also have value for potential users of the IMS data. With the exception of the the bias stability all other results are split diurnally. Therefore, we have updated the these result to also include the day night split:

[revised manuscript text omitted]

With these results we still observe some differences and similarities between the day and night results. However, 50% of the layers analysed are still within the trend uncertainty mainly due to the length of the time series. Therefore, we feel our final conclusion, that this is (to our knowledge) the first published looking at bias stability of IASI profiles and that we require longer time series to really investigate this properly. This is especially true when other IR sounders are processed with the IMS algorithm. Additional platforms with different sounding characteristics and overpass times will likely introduce other features that add to the noise, which in turn require longer time series for signal detection (Weatherhead et al. 1998). The new diurnal results point to a broader study where the effects which impact radiosonde biases also need to be accounted for in context satellite profile bias stability.